# LXRs link metabolism to inflammation through Abca1-dependent regulation of membrane composition and TLR signaling

Ayaka Ito[1], Cynthia Hong[1], Xin Rong[1], Xuewei Zhu[2,3], Elizabeth J Tarling[4], Per Niklas Hedde[5], Enrico Gratton[5], John Parks[2,3], Peter Tontonoz[6]*

[1]Department of Pathology and Laboratory Medicine, Howard Hughes Medical Institute, University of California, Los Angeles, Los Angeles, United States; [2]Department of Internal Medicine-Section on Molecular Medicine, Wake Forest School of Medicine, Winston-Salem, United States; [3]Department of Biochemistry, Wake Forest School of Medicine, Winston-Salem, United States; [4]Department of Medicine, University of California, Los Angeles, Los Angeles, United States; [5]Laboratory of Fluorescence Dynamics, Biomedical Engineering Department, Center for Complex Biological Systems, University of California, Irvine, Irvine, United States; [6]Howard Hughes Medical Institute, University of California, Los Angeles, Los Angeles, United States

**Abstract** The liver X receptors (LXRs) are transcriptional regulators of lipid homeostasis that also have potent anti-inflammatory effects. The molecular basis for their anti-inflammatory effects is incompletely understood, but has been proposed to involve the indirect tethering of LXRs to inflammatory gene promoters. Here we demonstrate that the ability of LXRs to repress inflammatory gene expression in cells and mice derives primarily from their ability to regulate lipid metabolism through transcriptional activation and can occur in the absence of SUMOylation. Moreover, we identify the putative lipid transporter Abca1 as a critical mediator of LXR's anti-inflammatory effects. Activation of LXR inhibits signaling from TLRs 2, 4 and 9 to their downstream NF-κB and MAPK effectors through Abca1-dependent changes in membrane lipid organization that disrupt the recruitment of MyD88 and TRAF6. These data suggest that a common mechanism-direct transcriptional activation-underlies the dual biological functions of LXRs in metabolism and inflammation.

*For correspondence:
ptontonoz@mednet.ucla.edu

Competing interests:
See page 21

## Introduction

The liver X receptors (LXRs) are members of the nuclear receptor superfamily that play pivotal roles in sterol homeostasis in mammals. LXRs control the expression of a battery of genes involved in cholesterol, fatty acid and phospholipid metabolism through direct binding to LXR response elements (LXREs) in their target promoters (*Calkin and Tontonoz, 2012*; *Hong and Tontonoz, 2014*). In addition to their ability to activate the expression of genes linked to lipid metabolism, LXRs also have the ability to antagonize inflammatory gene expression triggered by Toll-like receptor (TLR) activation (*Joseph et al., 2003*, *2004*; *Castrillo et al., 2003a*). LXRs are not known to act as direct ligand-dependent repressors, that is, there is no well-documented example of LXR/RXR heterodimers binding to an LXRE in a gene promoter and repressing transcription in response to ligand. Rather, they have been proposed to act on inflammatory promoters through mechanisms that do not involve DNA-binding domain recognition of LXREs.

**eLife digest** Inflammation is a normal part of the immune response to infection or tissue damage. However, increased inflammation has been linked to diseases such as obesity, diabetes and atherosclerosis (in which the walls of the arteries become hardened). These same diseases have also been linked to problems with the production or breakdown of fatty molecules, such as cholesterol.

Transcription factors are proteins that bind to DNA to control gene expression. A transcription factor called LXR regulates the production and breakdown of cholesterol in response to changing levels of cholesterol in the body. LXR has also been shown to inhibit inflammatory responses, but previous studies suggested that these two actions of LXR are independent of each other.

Ito et al. have now challenged these findings by showing that LXR inhibits inflammation via changes in the metabolism of cholesterol and other fatty molecules. The experiments used genetically engineered immune cells, called macrophages, and mice to show that activating LXR causes cholesterol molecules to move between the membranes in a cell. This in turn leads to changes in the signals sent by proteins found at the cell surface, and eventually to a reduction of inflammation responses.

Future work will focus on better understanding the link between LXR's effects on metabolism and inflammation in models of human diseases such as diabetes and atherosclerosis.

One proposed mechanism, termed 'transrepression', postulates that LXRs become SUMOylated in response to LXR agonist and that this SUMOylated LXR monomer stabilizes repressive nuclear complexes on the promoters of inflammatory genes. Key features of this model include the requirement for receptor SUMOylation, the involvement of an LXR monomer, and the mechanistic separation of the transcriptional activation and repression functions of LXRs (*Ghisletti et al., 2007*; *Lee et al., 2009*; *Venteclef et al., 2010*). Although several lines of evidence have supported the transrepression model, the requirement for receptor SUMOylation in the repressive actions of LXRs on inflammatory genes in cells and animals remains to be tested. The potential involvement of additional or alternative mechanisms has also not been excluded.

We demonstrate here that the ability of LXRs to antagonize endogenous inflammatory gene expression in cultured cells and in vivo is a consequence of changes in cellular lipid metabolism. We show that the ability of LXRs to activate transcription of the Abca1 sterol transporter, and thereby alter membrane cholesterol homeostasis, has a secondary effect on inflammatory signaling through inhibition of NF-κB and MAPK signaling pathways downstream of TLRs. These data present a unified view of LXR-dependent gene regulation in which direct transcriptional activation underlies the dual biological functions of LXRs in metabolism and inflammation. They further emphasize the importance of local membrane composition in the activation of TLR signaling pathways.

## Results

### Gene activation is required for inflammatory repression by LXRs

LXRs activate gene expression as heterodimers with RXRs. Previous studies have proposed that LXRs 'transrepress' inflammatory gene expression by acting as a monomer (*Ghisletti et al., 2007*; *Lee et al., 2009*; *Venteclef et al., 2010*). Unexpectedly, we found that siRNA-mediated knockdown of RXRα and RXRβ in primary mouse macrophages blunted the ability of LXR agonist to repress the LPS-induced expression of the endogenous inflammatory genes *Nos2, Il1β, Ccl2, Cxcl1, Tnfα, Cox2* and *Il6* (*Figure 1A* and *Figure 1—figure supplement 1A,B*). This observation suggested that LXR was acting to repress inflammation as a heterodimer with RXR in our system. We therefore explored alternative mechanisms for the repressive effects of LXR on inflammatory gene expression.

To test the structural requirements for LXR-dependent repression of endogenous inflammatory genes, we stably reconstituted immortalized mouse embryonic fibroblasts (MEFs) and immortalized primary bone marrow-derived macrophages (iBMDM) from mice lacking LXRα and LXRβ with wild-type and mutant LXRs (*Figure 1, Figure 1—figure supplement 1*). The synthetic LXR agonist GW3965 did not induce the canonical LXR target gene *Abca1*, nor did it repress LPS-induced inflammatory

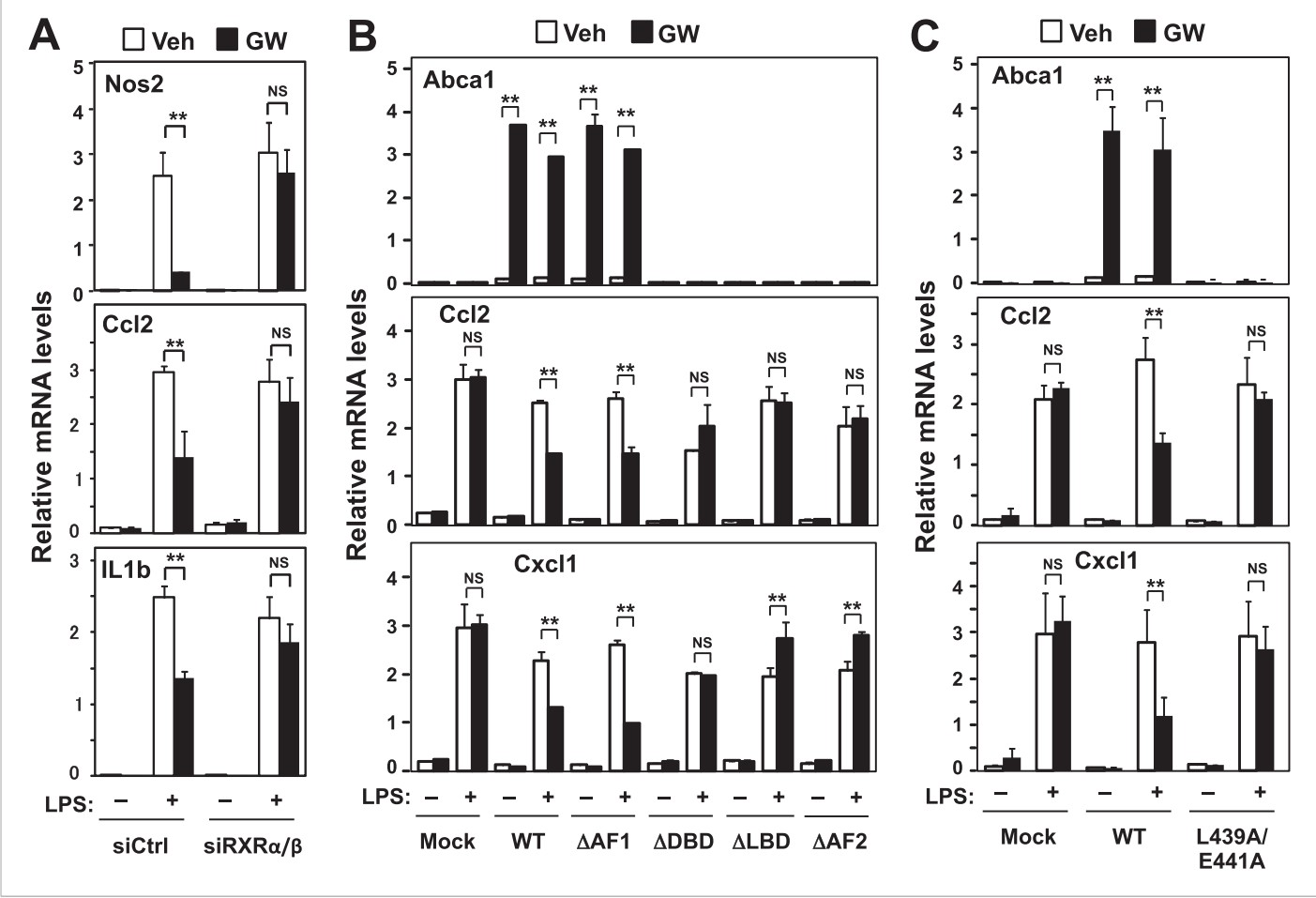

**Figure 1**. RXR and transactivation are required for LXR-dependent inflammatory repression. (**A**) Bone marrow-derived macrophages from wild-type mice were transfected with siRNA targeting RXRα and RXRβ (siRXRαβ or control (siCtrl) for 48 hr, pretreated with GW3965 (1 µM) overnight, and then stimulated with LPS (10 ng/ml) for 4 hr. (**B**) Immortalized MEFs from *Lxrα−/−Lxrβ−/−* mice reconstituted with wild-type human LXRα, AF1-deletion mutant (ΔAF1), DNA-binding domain deletion mutant (ΔDBD), ligand-binging domain deletion mutant (ΔLBD), AF2-deletion mutant (ΔAF2) or control mock were pretreated with the LXR agonist GW3965 (1 µM) overnight, followed by stimulation with LPS (10 ng/ml) for 4 hr. (**C**) Immortalized MEFs from *Lxrα−/−Lxrβ−/−* mice reconstituted with wild-type human LXRα, L439A/E441A mutant or control mock were pretreated with the LXR agonist GW3965 (1 µM) overnight, followed by stimulation with LPS (10 ng/ml) for 4 hr. Gene expression was analyzed by real-time PCR. N = 4 per group. *p < 0.05, **p < 0.01, NS, not significant. Error bars represent means ± SEM.

The following figure supplement is available for figure 1:

**Figure supplement 1**. Effects of RXR knockdown on LXR-mediated inflammatory repression.

gene expression, in LXR-deficient MEFs or macrophages (*Figures 1B,C, 2A,B*) (*Castrillo et al., 2003a*, *2003b*; *Joseph et al., 2004*). However, when LXR-deficient cells were reconstituted with wild-type LXRα, GW3965 treatment simultaneously induced *Abca1* expression and repressed LPS-induced inflammatory gene expression (*Ccl2* and *Cxcl1*) (*Figure 1B,C*). These results establish that both the activating and repressive effects of GW3965 under our experimental conditions are entirely mediated by LXRs. This is important to emphasize because many synthetic LXR agonists, including GW3965 and T0901317, can exhibit LXR-independent effects on gene expression, especially when used at concentrations higher than employed here. It is also important to note that this approach does not involve supraphysiologic overexpression of LXRs, as the level of reconstituted protein expression in these cells was within the physiologic range and restored target gene expression to physiologic levels (*Figure 1—figure supplement 1C,D*).

To identify the domains important for LXR-dependent repression, we reconstituted LXR-deficient cells with various domain-deletion mutants of LXR. Agonist treatment induced *Abca1* and repressed the levels of *Ccl2* and *Cxcl1* in AF1-deletion mutant cells. However, both the activation and repression activities of LXR were completely abolished in DNA binding domain- (DBD), ligand binding domain- (LBD) or AF2-deletion mutant cells (*Figure 1B*). Thus, the DBD, LBD and AF2 domains, but not the AF1 domain, are critical for both transactivation and repression by LXR. Furthermore, an LXR mutant defective in its ability to recruit co-activators, L439A/E441A (*Tzukerman et al., 1994*; *Bastie et al., 2000*), was unable to induce *Abca1* or to repress inflammatory genes in both MEFs and iBMDM (*Figure 1C*, *Figure 2B*, *Figure 2—figure supplement 1A*), strongly suggesting that gene activation and inflammatory repression are mechanistically linked.

In contrast to activation-defective LXR proteins, mutants of LXRα and LXRβ that lack the SUMOylation sites previously identified (*Ghisletti et al., 2007*) were capable of inducing *Abca1* and repressing inflammatory genes in both MEFs and iBMDM (*Figure 2A,B* and *Figure 2—figure supplement 1A*). We considered the possibility that alternative SUMOylation sites might be employed, however, mutation of three additional residues in the hinge region of the receptor predicted to be likely SUMOylation sites also had no effect on LXR-dependent repression of *Cxcl1* or *Ccl2* (*Figure 2—figure supplement 1B*). We also considered the possibility that SUMOylation might be required for a distinct subset of inflammatory genes in our system. However, transcriptional profiling revealed that LXR SUMOylation site mutants were capable of mediating ligand-dependent repression of a broad array of inflammatory genes (e.g., those annotated with the 'Immune system process' GO term; *Figure 2C*).

We further tested whether the SUMOylation machinery reported to target LXR was required for inflammatory repression in our system. Knocking down Ubc9 or Hdac4 by siRNA in iBMDM did not block repression (*Figure 2—figure supplement 2*). Thus, SUMOylation-dependent transrepression could not account for the inhibitory actions of LXR on inflammatory gene expression in our system.

## Abca1 induction contributes to LXR-mediated inflammatory repression

Given our data suggesting that gene activation and inflammatory repression by LXRs were mechanistically linked, we hypothesized that there must be one or more direct LXR target genes whose action has a secondary effect on inflammatory gene expression. We therefore used siRNA to knockdown a panel of LXR target genes in primary BMDM and tested whether inflammatory repression was compromised. Surprisingly, we found that knockdown of *Abca1*, a gene critical for cellular cholesterol efflux (*Oram, 2003*), substantially impaired the ability of LXR agonists to repress inflammatory gene expression (*Figure 3A*). By contrast, knockdown of *Abcg1* or *Apoe*, two other LXR target genes involved in metabolism, had no effect on inflammatory repression (*Figure 3A* and *Figure 3—figure supplement 1*).

We corroborated these results by analyzing the effects of LXR agonist in BMDM from myeloid-specific *Abca1−/−* mice. Again, the ability of LXR agonist to repress inflammation was markedly inhibited in the genetic absence of *Abca1* (*Figure 3B*), but not *Abcg1* (*Figure 3—figure supplement 1A*). To comprehensively analyze the impact of *Abca1* deficiency on inflammatory gene expression, we performed transcriptional profiling. This analysis revealed that most of the genes annotated with the 'Immune system process' GO term were not repressed by LXR activation in *Abca1*-deficient macrophages (*Figure 3C*), suggesting that LXR-induced *Abca1* expression is broadly important for LXR-mediated repression. We also tested whether Abca1 was required for inflammatory repression by the synthetic glucocorticoid receptor (GR) agonist dexamethasone. The ability of dexamethasone to repress inflammatory gene expression was preserved in *Abca1−/−* cells, indicating that distinct mechanisms are involved in transcriptional repression by LXR and GR (*Figure 3—figure supplement 1C*).

Prior work has shown that LXR-dependent induction of *Abca1* contributes to the maintenance of sterol homeostasis by promoting the efflux of cellular cholesterol to extracellular acceptors such as ApoA-I and ApoE (*Santamarina-Fojo et al., 2001*; *Oram, 2003*). Tangier disease is a severe HDL deficiency syndrome that is caused by mutations in the *ABCA1* gene. We also tested the ability of LXR agonist to inhibit inflammatory gene expression in skin fibroblasts from a patient with Tangier disease that lack functional ABCA1 protein. LXR target genes were induced at similar levels in normal (healthy donor) and Tangier fibroblasts in response to activation by synthetic agonist (*Figure 4A*). While LXR activation with ligand repressed inflammatory gene expression in normal cells, this response was abrogated in Tangier fibroblasts (*Figure 4B*).

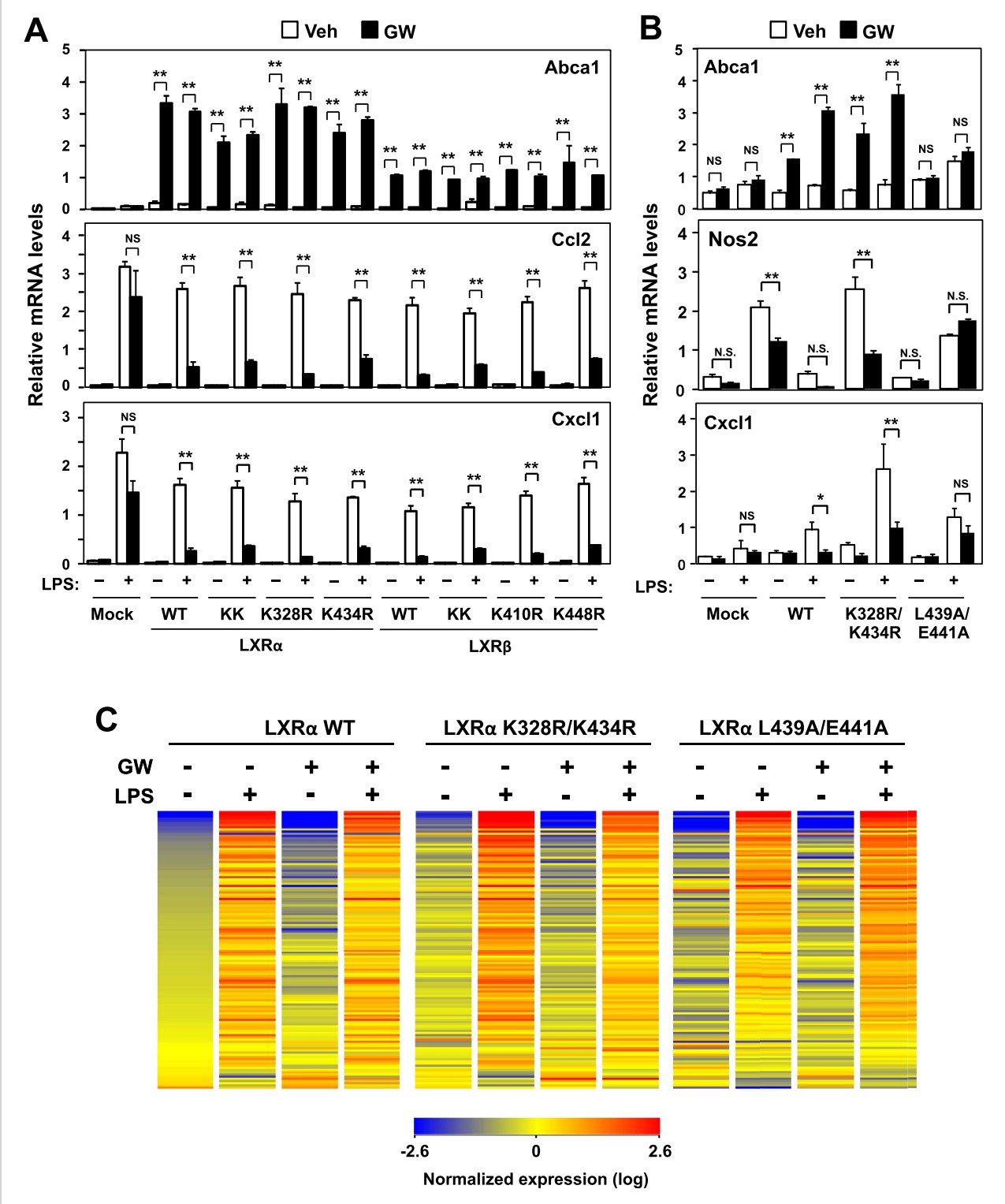

**Figure 2**. Transactivation but not sumoylation is required for LXR-mediated inflammatory repression. (**A**) Immortalized MEFs from *Lxrα−/−Lxrβ−/−* mice reconstituted with wild-type human LXRα, sumoylation site mutants (K328R/K434R (KK), K328R, K434R), wild-type human LXRβ, sumoylation site mutants (K410R/K448R (KK), K410R, K448R), or mock control were pretreated with GW3965 (1 μM) overnight, followed by stimulation with LPS (10 ng/ml) for 4 hr. (**B**, **C**) Immortalized bone marrow-derived macrophages from *Lxrα−/−Lxrβ−/−* mice reconstituted with wild-type human LXRα, K328R/K434R (KK) mutant, L439A/E441A mutant, or mock control were pretreated with GW3965 (1 μM) overnight, followed by stimulation with LPS (10 ng/ml) for 4 hr. Gene expression was analyzed by real-time PCR (**B**) and Agilent microarrays (**C**). Selected genes that are annotated with the 'Immune system process' GO
*Figure 2. continued on next page*

Figure 2. Continued

term from the array studies are presented as a heatmap (≧ twofold change). N = 4 per group. *p < 0.05, **p < 0.01, NS, not significant. Error bars represent means ± SEM.

The following figure supplements are available for figure 2:

**Figure supplement 1**. Effect of lysine mutants on LXR-mediated inflammatory repression.

**Figure supplement 2**. Repression does not require Ubc9 or Hdac4.

To further examine the function of Abca1 was critical for inflammatory repression, we reconstituted iBMDM from myeloid-specific *Abca1−/−* mice with wild-type Abca1 or two different Abca1 point mutants that lacks cholesterol efflux ability, N935S and C1477R (*Singaraja et al., 2006*; *Kannenberg et al., 2013*). When *Abca1*-deficient cells were reconstituted with wild-type Abca1, inflammatory gene expression was repressed by LXR activation; however, this effect was lost in cells expressing the N935S or C1477R mutants (*Figure 4C*). These data suggest that cholesterol transport by Abca1 is critical for inflammatory repression in human cells and that effective LXR-dependent inhibition of inflammation requires the ATP-dependent transporter function of Abca1.

## Cellular cholesterol content affects inflammatory responses

The requirement for Abca1 in LXR-dependent inflammatory repression suggests that membrane cholesterol content or distribution is mechanistically linked to inflammatory gene repression. To test this idea, we loaded cells with cholesterol using cyclodexrin-cholesterol complexes or depleted cells of cholesterol by incubating them with hydroxypropyl-β-cyclodextrin (*Klein et al., 1995*; *Christian et al., 1997*). As expected, cholesterol addition activated LXR target genes and repressed SREBP target genes, whereas cholesterol removal inhibited LXR target genes and induced SREBP target genes (*Figure 5*). Remarkably, increasing the cholesterol content of the cell also enhanced the induction of inflammatory genes by LPS. On the other hand, decreasing cholesterol content inhibited the induction of inflammatory genes (*Figure 5*). These data are consistent with the hypothesis that regulation of membrane cholesterol content by LXRs may modulate inflammatory responses.

## LXR activation inhibits inflammatory signaling through Abca1 induction

To further explore the connection between LXR-dependent target gene activation and inflammatory repression, we analyzed signaling pathways downstream of TLR4 in macrophages. We found that treatment of macrophages with LXR agonist inhibited the activation of ERK, p38 and JNK MAP kinases downstream of TLR4 activation (*Figure 6A,B*). Reduced levels of phospho-ERK, phospho-p38 and phospho-JNK were observed in the setting of LXR activation. Importantly, this effect was mediated by LXRs, as it was not observed in LXR-deficient macrophages. To test whether inhibition of MAPK signaling was mechanistically linked to the regulation of *Abca1* expression by LXR, we repeated these studies in macrophages derived from myeloid-specific *Abca1*-deficient mice. Remarkably, the ability of LXR agonist to inhibit ERK, p38 and JNK activation was severely impaired in cells lacking *Abca1* (*Figure 6C,D*).

We also found that LXR agonist suppressed the activation of NF-κB signaling downstream of TLR4. Immunoblot analysis of whole cell lysates and nuclear extracts revealed that treatment of macrophages with GW3965 reduced the ratio of phospho-IκBα to total IκBα and reduced the nuclear abundance of p65 (*Figure 7A,B*). These effects of agonist were also LXR-dependent, as they were not observed in LXR-deficient macrophages. Furthermore, the ability of LXR agonist to suppress NF-κB signaling was abolished in *Abca1*-deficient macrophages (*Figure 7C,D*). Analysis of p65 localization by confocal immunofluorescence microscopy confirmed the ability of GW3965 to inhibit p65 nuclear translocation in an LXR-dependent and Abca1-dependent manner (*Figure 7E,F*).

To address whether p65-binding to its target inflammatory gene promoters was affected by LXR activation, we employed chromatin immunoprecipitation (ChIP) assays. The binding of p65 to the *Nos2*, *IL1b* and *Ccl2* gene promoters was increased by LPS treatment as expected, whereas no binding was observed on the hemoglobin beta (*Hbb2*) gene promoter, which does not contain a p65 binding site (*Figure 8*). We found that LXR activation suppressed LPS-induced p65 recruitment to the

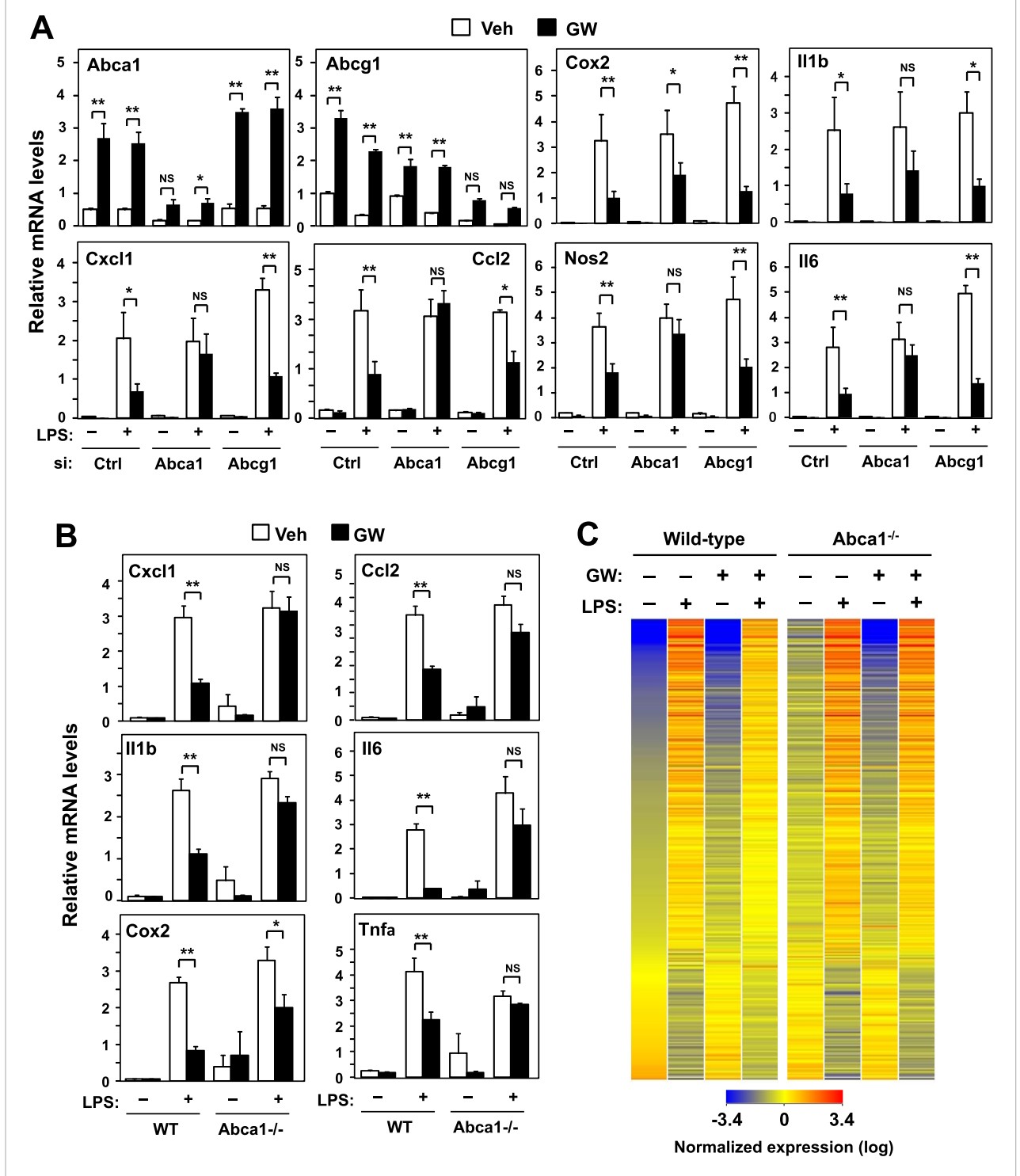

**Figure 3**. ABCA1 induction is critical for LXR-mediated repression. (**A**) Bone marrow-derived macrophages from wild-type mice were transfected with siRNA targeting Abca1, Abcg1 or control (Ctrl) for 48 hr, pretreated with GW3965 (1 µM) overnight, and then stimulated with LPS (10 ng/ml) for 4 hr. (**B**, **C**) Bone marrow-derived macrophages from myeloid-specific *Abca1−/−* and control wild-type mice were pretreated with GW3965 (1 µM) overnight, followed by stimulation with LPS (10 ng/ml) for 4 hr. Gene expression was analyzed by real-time PCR (**B**) and Agilent microarrays (**C**). Selected genes from the array studies that are annotated with the 'Immune system process' GO term are presented as a heatmap (≥twofold changes shown). N = 4 per group. *p < 0.05, **p < 0.01, NS, not significant. Error bars represent means ± SEM.

*Figure 3. continued on next page*

*Figure 3. Continued*

The following figure supplement is available for figure 3:

**Figure supplement 1**. Loss of Abcg1 or ApoE does not compromise LXR-mediated repression.

*Nos2s, Ccl2* and *Il1b* gene promoters. Moreover, the inhibitory effect of LXR agonist on p65 recruitment was lost in Abca1-deficient macrophages (*Figure 8*). Together, the data of *Figures 6–8* demonstrate that LXR activation inhibits signaling through TLR4, and that Abca1 expression is critical for this phenomenon.

## LXR-dependent Abca1 expression redistributes membrane cholesterol and disrupts TLR signaling complexes in lipid rafts

We next tested the hypothesis that LXR agonist treatment relocalizes plasma membrane cholesterol away from detergent-resistant microdomains or rafts, thereby preventing the localization of TLR4 signaling molecules and impairing downstream signaling. BMDM from wild-type or LXR-deficient mice were treated overnight with GW3965, and then cell membranes were fractionated and the cholesterol content of each fraction determined. We found that LXR activation reduced the cholesterol content of Flotillin-enriched detergent-resistant membrane microdomains in wild-type type (<47% reduction) but not LXR-deficient macrophages (*Figure 9A*). Furthermore, this reduction in cholesterol content of detergent-resistant membrane microdomains was not observed in Abca1-deficient macrophages (*Figure 9B*). On the other hand, the expression of ganglioside GM1, a marker of lipid rafts, was not altered by LXR agonist (*Figure 9C*). These results suggest that LXR-dependent ABCA1 expression reduces the cholesterol content of detergent-resistant membrane microdomains without affecting the abundance of lipid rafts in the plasma membrane.

The cholesterol content of the plasma membrane affects its fluidity and rigidity, and therefore could conceivably affects signaling through membrane receptors (*Simons and Toomre, 2000*; *Fessler and Parks, 2011*). Laurdan is a fluorescent lipophilic molecule that can be used to detect changes in membrane dynamics due to its sensitivity to the polarity of the membrane environment (*Parasassi and Gratton, 1995*; *Vest et al., 2006*; *Golfetto et al., 2013*). Changes in membrane dynamics shift the laurdan emission spectrum, which can be quantified by the generalized polarization (GP) calculated from the spectrum shifts (*Parasassi et al., 1990*). To examine if membrane dynamics were affected as a result of LXR-dependent reductions in raft cholesterol content, we stained living primary macrophages with laurdan. As shown in *Figure 9D*, plasma membrane rigidity was decreased in response to LXR activation, consistent with the altered cholesterol distribution. Furthermore, the ability of LXR agonist treatment to affect plasma membrane fluidity in primary macrophages was largely dependent on Abca1expression, as we did not observe a shift in laurdan signal in Abca1−/− macrophages (*Figure 9D*).

To gain additional insight into the inflammatory signaling pathways impacted by LXR, we tested the influence of LXR agonist treatment on gene expression induced by TLR2 (Pam3CSK4), TLR9 (CpG) and TLR3 (Poly I:C) agonists. LXR activation blunted the response to TLR2 and TLR9 (*Figure 10A,B*), but did not affect the induction of viral response genes by TLR3 (*Figure 10C*). Furthermore, loss of Abca1 expression in macrophages inhibited the ability of LXR agonist to repress gene expression stimulated by TLR2 and TLR9 as well as by TLR4. These observations suggested that LXR-dependent membrane lipid remodeling was targeting a component of the inflammatory signaling cascade common to TLRs 2, 4 and 9.

The adaptor molecule MyD88 is commonly engaged by TLRs 2, 4 and 9 and links these receptors to the downstream activation of MAPKs and NF-κB, and ultimately to the induction of cytokine gene expression. Thus, MyD88 emerged as potential common component of inflammatory signaling pathways targeted by LXR and Abca1. Flow cytometry revealed that TLR4 expression in plasma membrane was not affected by LXR activation (*Figure 11A*). We therefore tested whether the recruitment of key adaptor molecules to lipid rafts was altered by LXR activation. We performed cell fractionation studies to assess the ability of MyD88 and TRAF6 to associate with lipid rafts in response to TLR4 activation by LPS. We confirmed the ability of our antibodies to detect MyD88 and TRAF6 using MyD88-deficient and siRNA to TRAF6 (*Figure 11—figure supplement 1*). LPS-dependent recruitment of MyD88 and TRAF6 in Flotillin-1-enriched detergent-resistant membrane microdomains

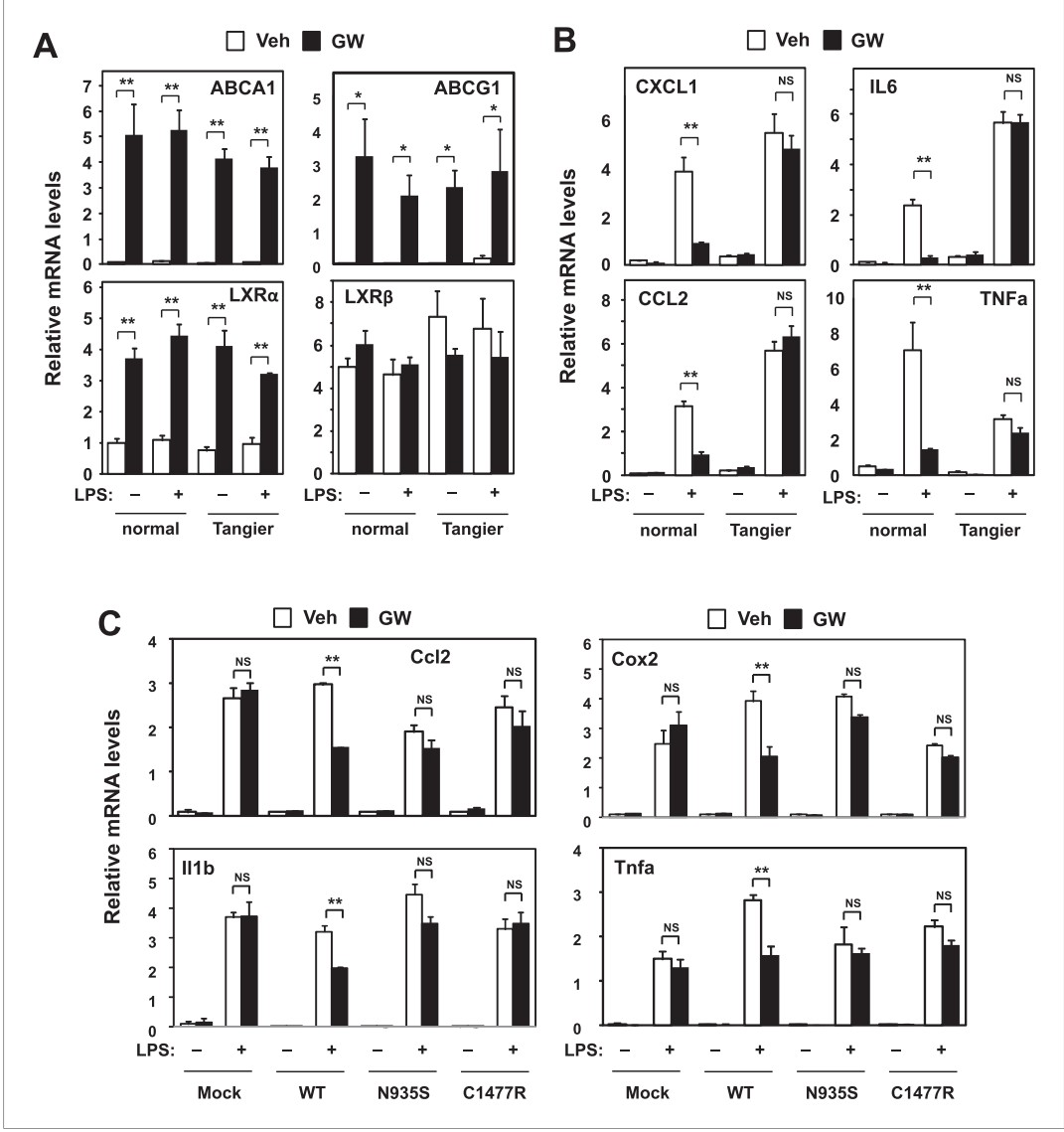

**Figure 4**. Intracellular cholesterol content affects LXR-mediated repression. (**A**, **B**) Skin fibroblasts from a healthy donor (normal) and a Tangier disease patient (Tangier) were pretreated with GW3965 (1 µM) overnight, followed by stimulation with LPS (10 ng/ml) for 4 hr. (**C**) Immortalized bone marrow-derived macrophages from *Abca1−/−* mice reconstituted with wild-type Abca1, N935S mutant, C1447R mutant or mock control were pretreated with GW3965 (1 µM) overnight, followed by stimulation with LPS (10 ng/ml) for 4 hr. Gene expression was analyzed by real-time PCR. N = 4 per group. *p < 0.05, **p < 0.01, NS, not significant. Error bars represent means ± SEM.

was markedly reduced by treatment of iBMDM with GW3965 (*Figure 11B*). Furthermore, the ability of LXR agonist to inhibit recruitment of MyD88 and TRAF6 was lost in Abca1-deficient macrophages. These observations suggest that a reduction in raft cholesterol content in response to LXR activation and Abca1 expression leads to alterations in membrane dynamics and/or raft structure and to the disruption of functional TLR4 complexes. Therefore, downstream MAPK and NF-κB signaling and inflammatory gene expression are repressed.

## Abca1 contributes to LXR-dependent repression of inflammation in vivo

To test whether Abca1 was required for the ability of LXR agonists to repress inflammatory gene expression in vivo, we pretreated wild-type or myeloid-specific *Abca1−/−* mice with vehicle or GW3965 for 3 days and then challenged them with LPS. Analysis of gene expression in spleen and

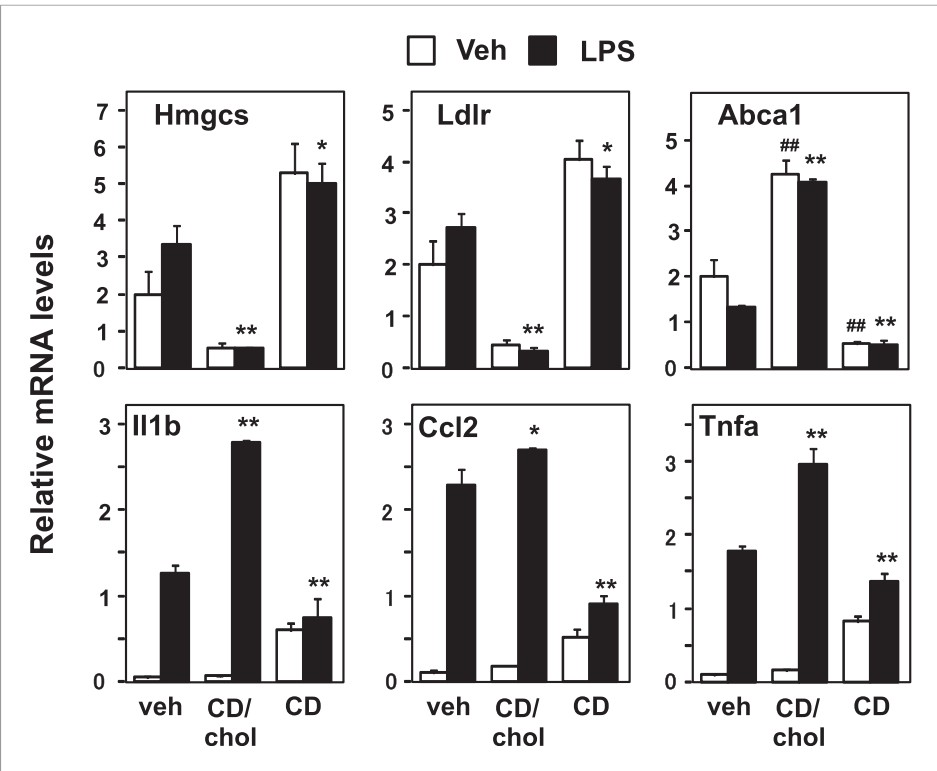

**Figure 5**. Manipulation of membrane cholesterol content affects inflammatory responses. Bone marrow-derived macrophages were incubated with cyclodextrin cholesterol (CD-Chol, 100 μM) or hydroxypropyl-β-cyclodextrin (CD, 10 mM) for 1 hr and stimulated with LPS (10 ng/ml) for 4 hr. Gene expression was analyzed by real-time PCR. N = 4 per group. *$p < 0.05$, **$p < 0.01$, NS, not significant. Error bars represent means ± SEM.

lung 2 hr after LPS treatment revealed that an array of inflammatory genes was induced by LPS as expected (*Figure 12*). This induction was substantially attenuated by LXR agonist treatment in wild-type mice, whereas little if any inflammatory repression was observed in mice lacking expression of Abca1 in macrophages (*Figure 12*). Taken together, these results suggest that the ability of LXRs to repress inflammation in macrophages derives primarily from their ability to regulate cellular cholesterol metabolism through Abca1. They further reveal an unexpected connection between Abca1-dependent membrane cholesterol distribution and TLR4-mediated inflammatory signaling.

## Discussion

LXRs reciprocally regulate lipid metabolism and inflammation. Both of these processes are central to the pathogenesis of metabolic diseases such as atherosclerosis and diabetes (*Castrillo et al., 2003a*, *2003b*; *Joseph et al., 2004*). LXRs have been proposed to repress inflammatory gene expression *in trans* by tethering to NF-κB transcriptional complexes in a SUMOylation-dependent manner (*Ghisletti et al., 2007*; *Lee et al., 2009*; *Venteclef et al., 2010*). This has heretofore been believed to be the sole mechanism underlying LXR's anti-inflammatory effects. The present study suggests that LXR-mediated transcriptional repression of endogenous inflammatory genes is principally dependent on LXR's capacity for transactivation. Our studies also identify the sterol transporter Abca1 as an important mediator of LXR's anti-inflammatory effects, both in vitro and in vivo. We find that Abca1 induction represses TLR-induced NF-κB and MAPK pathway activation by reducing the cholesterol content of detergent-resistant membrane domains. These findings reveal a previously unrecognized mechanism for LXR-mediated inflammatory repression in which inflammatory repression is secondary to activation-dependent changes in cellular lipid metabolism.

Although we cannot exclude the possibility that trans-repression may operate in certain contexts, LXR agonist did not repress endogenous inflammatory gene expression in the absence of RXRα and

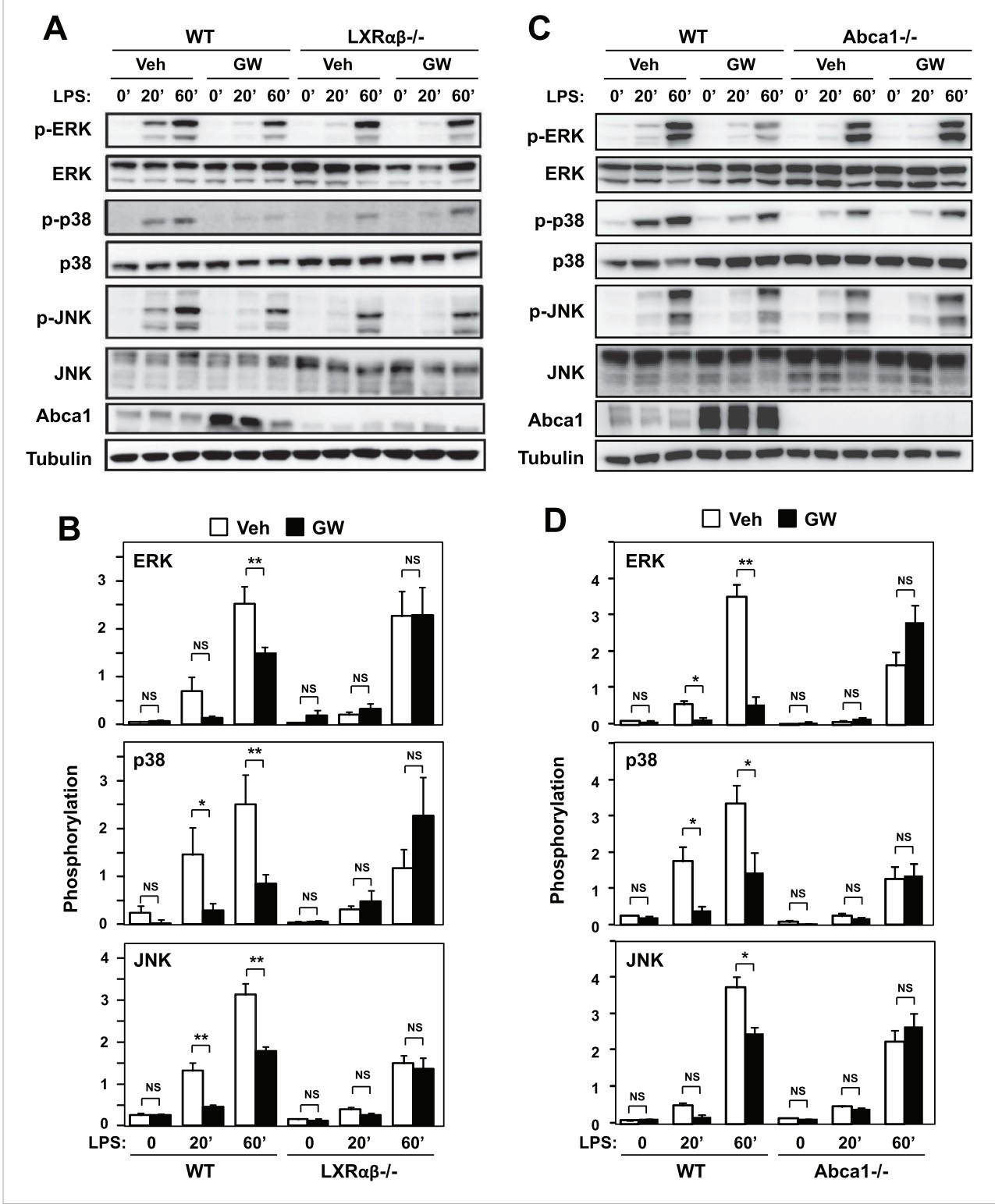

**Figure 6**. Ligand activation of LXR inhibits LPS-induced MAP kinase activation through Abca1 induction. (**A**–**D**) Bone marrow-derived macrophages from *Lxrα−/−Lxrβ−/−* and control wild-type mice (**A**, **B**), or bone marrow-derived macrophages from myeloid-specific *Abca1−/−* and control wild-type mice (**C**, **D**) were pretreated with GW3965 (1 µM) overnight, followed by stimulation with LPS (10 ng/ml) for 20 min or 1 hr. Whole cell lysates were harvested and protein expression was analyzed by immunoblotting with the indicated antibodies (**A**, **C**). Protein expression was quantified by Image Quant TL7.0 (**B**, **D**). N = 4–6 per group. *p < 0.05, **p < 0.01, NS, not significant. Error bars represent means ± SEM.

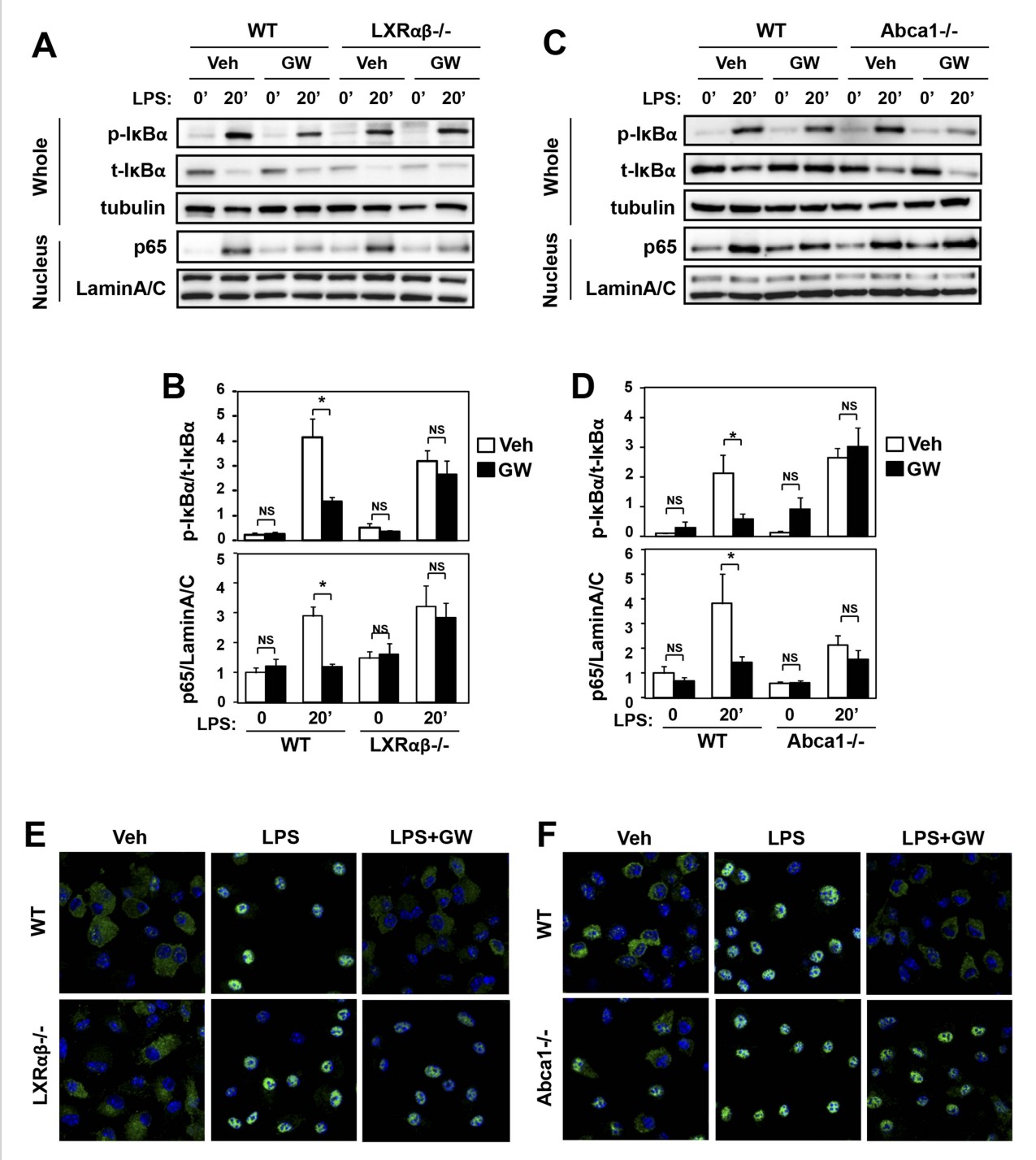

**Figure 7**. Ligand activation of LXR inhibits LPS-induced NF-κB activation through Abca1 induction. (**A–F**) Bone marrow-derived macrophages from *Lxrα–/–Lxrβ–/–* and control wild-type mice (**A**, **B**, **E**), or bone marrow-derived macrophages from myeloid-specific *Abca1–/–* and control wild-type mice (**C**, **D**, **F**) were pretreated with GW3965 (1 μM) overnight, followed by stimulation with LPS (10 ng/ml) for 20 min. Whole cell lysates and nuclear lysates were harvested and protein expression was analyzed by immunoblotting with the indicated antibodies (**A**, **C**). Protein expression was quantified by Image Quant TL7.0 (**B**, **D**). N = 4–6 per group. *p < 0.05, **p < 0.01, NS, not significant. Error bars represent means ± SEM. Nuclear translocation of p65 was assessed by staining of p65 (green) and DAPI (blue) (**E**, **F**).

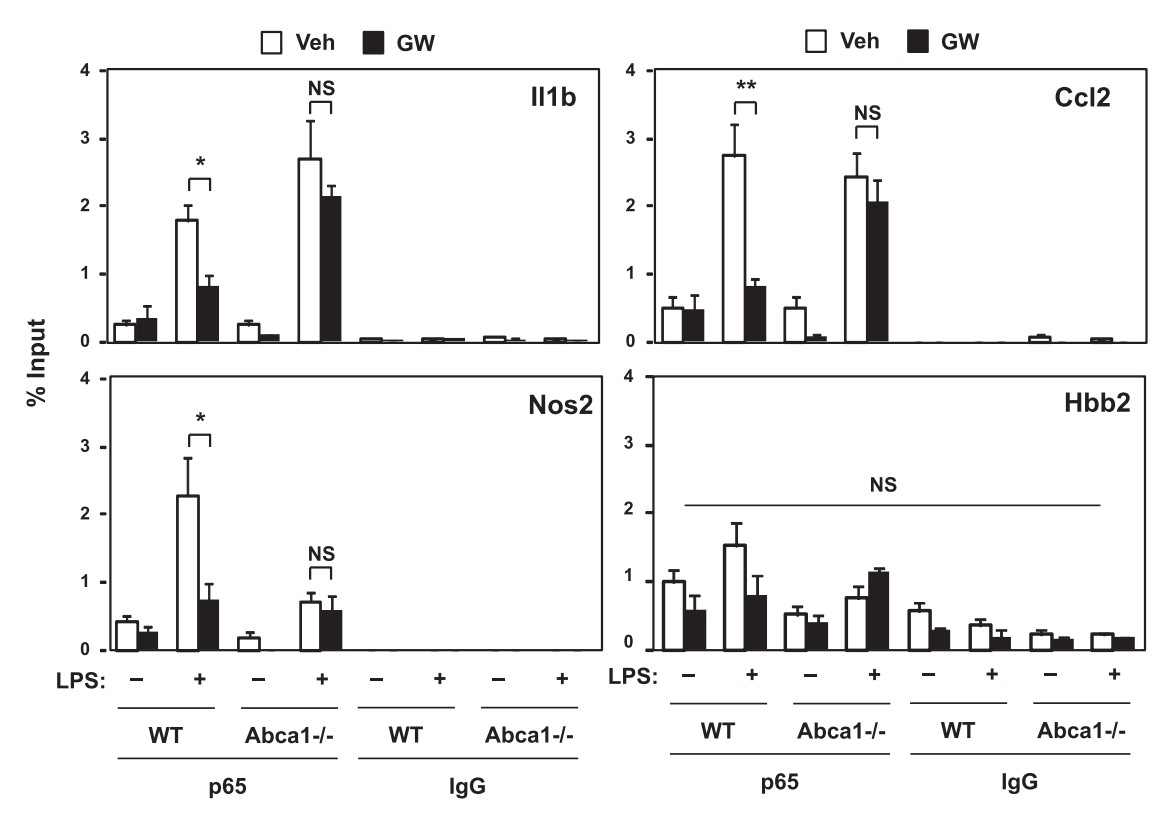

**Figure 8**. LXR activation decreases p65 occupancy on inflammatory gene promoters. Recruitment of p65 to inflammatory gene promoters was assessed by ChIP-qPCR assays. Chromatin from wild-type or *Abca1−/−* cells was precipitated with p65 antibody or control IgG. N = 4 per group. *p < 0.05, **p < 0.01, NS, not significant. Error bars represent means ± SEM.

RXRβ in our system, suggesting that LXR/RXR heterodimers were mediating repression. This led us to explore additional mechanisms for LXR inflammatory crosstalk. We found that an LXR L439A/E441A mutant that lacks co-activator recruitment capacity was unable to repress inflammation. These studies suggested that repression was mechanistically linked with transcriptional activation, and that repression might be a secondary phenomenon.

We examined an array of endogenous gene expression in LXR-null cells reconstituted with various LXRs mutants and found that mutants that could not undergo SUMOylation at the previously identified lysines (LXRα K328 and K434 and LXRβ K410 and K448) were still competent to repress inflammatory gene expression. Interestingly, the nuclear receptor SF-1 can be modified by SUMO in its hinge region and this modification has been shown to be functionally important for gene repression in vivo (*Campbell et al., 2008*; *Lee et al., 2011*). In contrast to the LXR transrepression model, however, the repressive effects of SUMOylated SF-1 are mediated by direct binding of SF-1 to response elements in its target promoters. Nevertheless, we considered the possibility that alternative SUMOylation sites in the hinge region of LXR (LXRα K177/K178/K180) might be used for transrepression. However, mutation of the three hinge lysines to arginine did not alter LXR-dependent repression of endogenous inflammatory genes.

Prior studies from our group suggested that LXR-mediated inflammatory repression was not mediated by the inhibition of nuclear translocation or DNA binding of NF-κB and AP-1 (*Castrillo et al., 2003a*, *2003b*). These conclusions were drawn based on the analysis of electrophoretic mobility shift assays. In retrospect, these studies likely had technical limitations that prevented us from appreciating the effects of LXR on NF-κB signaling. Our prior studies did not define which NF-κB isoforms were bound to our DNA probe, and according to a recent comprehensive analysis of DNA binding by NF-κB proteins (*Siggers et al., 2012*), the sequence we used is not predicted to have high

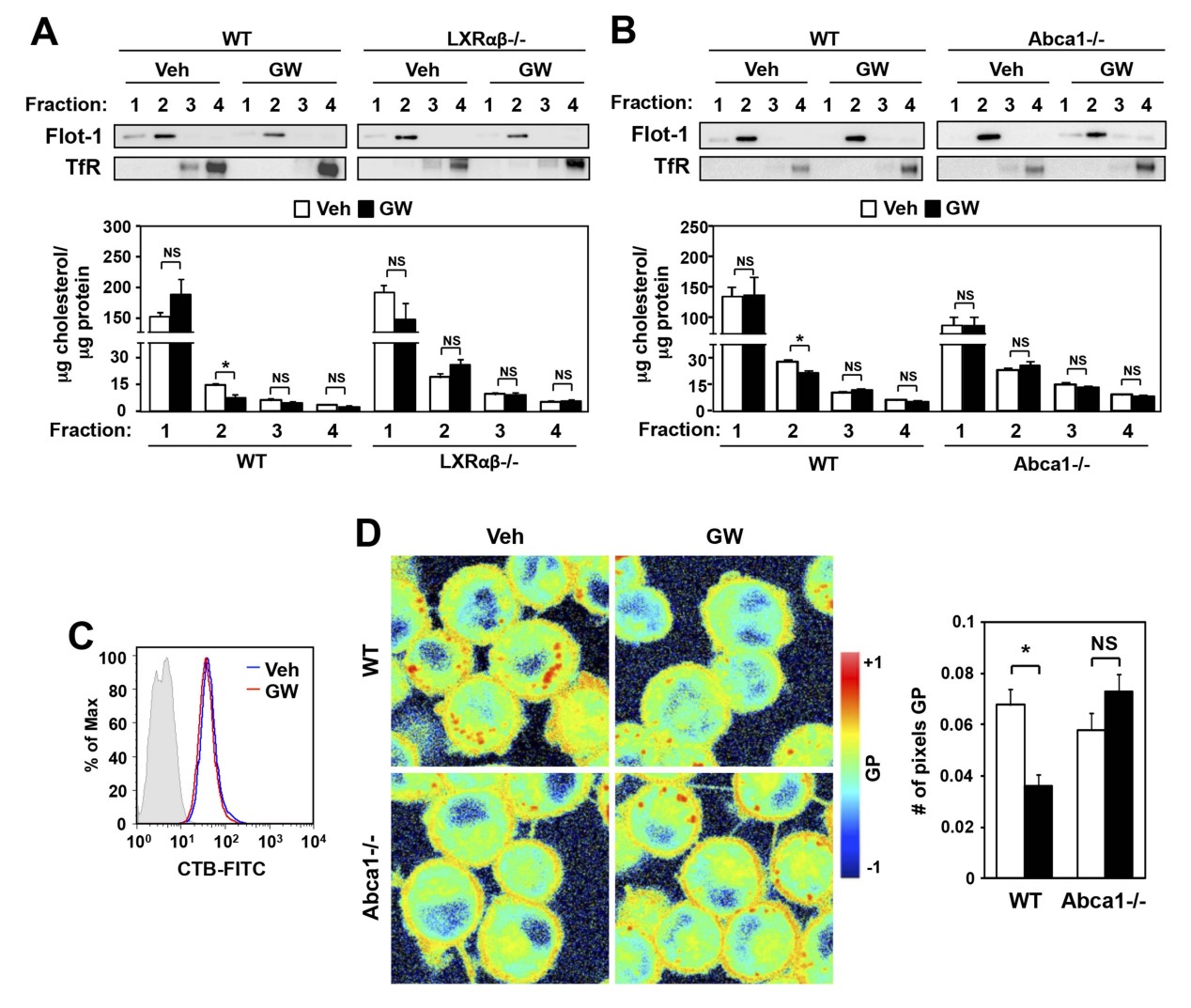

**Figure 9**. Ligand activation of LXR decreases cholesterol content in lipid rafts and increases membrane lipid mobility. (**A**, **B**) Bone marrow-derived macrophages from *Lxrα−/−Lxrβ−/−* and control wild-type mice (**A**), or bone marrow-derived macrophages from myeloid-specific *Abca1−/−* and control wild-type mice (**B**) were pretreated with GW3965 (1 μM) overnight. Lipid raft and non-raft fractions were isolated using the detergent method and each fraction was analyzed by Western blotting. The free cholesterol concentration in each fraction was determined and normalized to protein concentration. N = 5 per group. (**C**) The abundance of lipid rafts in the plasma membrane was analyzed by flow cytometry after staining with cholera toxin B (CTB). (**D**) Wild-type or *Abca1−/−* iBMDM were treated with oxidized LDL (50 μg/ml) for 48 hr, and GW3965 (1 μM) overnight. GP images (left) were obtained from fluorescence-lifetime imaging microscopy (FLIM). The GP scale used to pseudocolor the intensity image is shown at right. Plasma membrane fluidity was determined with GP and number of pixel in plasma membrane (right). *p < 0.05, **p < 0.01, NS, not significant. Error bars represent means ± SEM.

affinity for the LPS-induced p65-p50 heterodimer (*Hoffmann and Baltimore, 2006*). Our subsequent analysis of IκB phosphorylation, p65 nuclear translocation, and p65 recruitment to gene promoters has revealed previously unrecognized impairments in NF-κB and MAPK signaling in the setting of LXR activation.

The observation that gene activation was required for LXR-mediated inflammatory repression led us to identify the cholesterol transport protein Abca1 as a key mediator of the phenomenon. Accumulating evidence suggests that changes in membrane lipid content can affect inflammatory responses (*Westerterp et al., 2013*; *Rong et al., 2015*). Our membrane dynamics and biochemical studies indicate that LXR-dependent Abca1 induction reduces raft cholesterol content and alters membrane fluidity, suggesting a biophysical basis for the ability of LXR to inhibit TLR signaling.

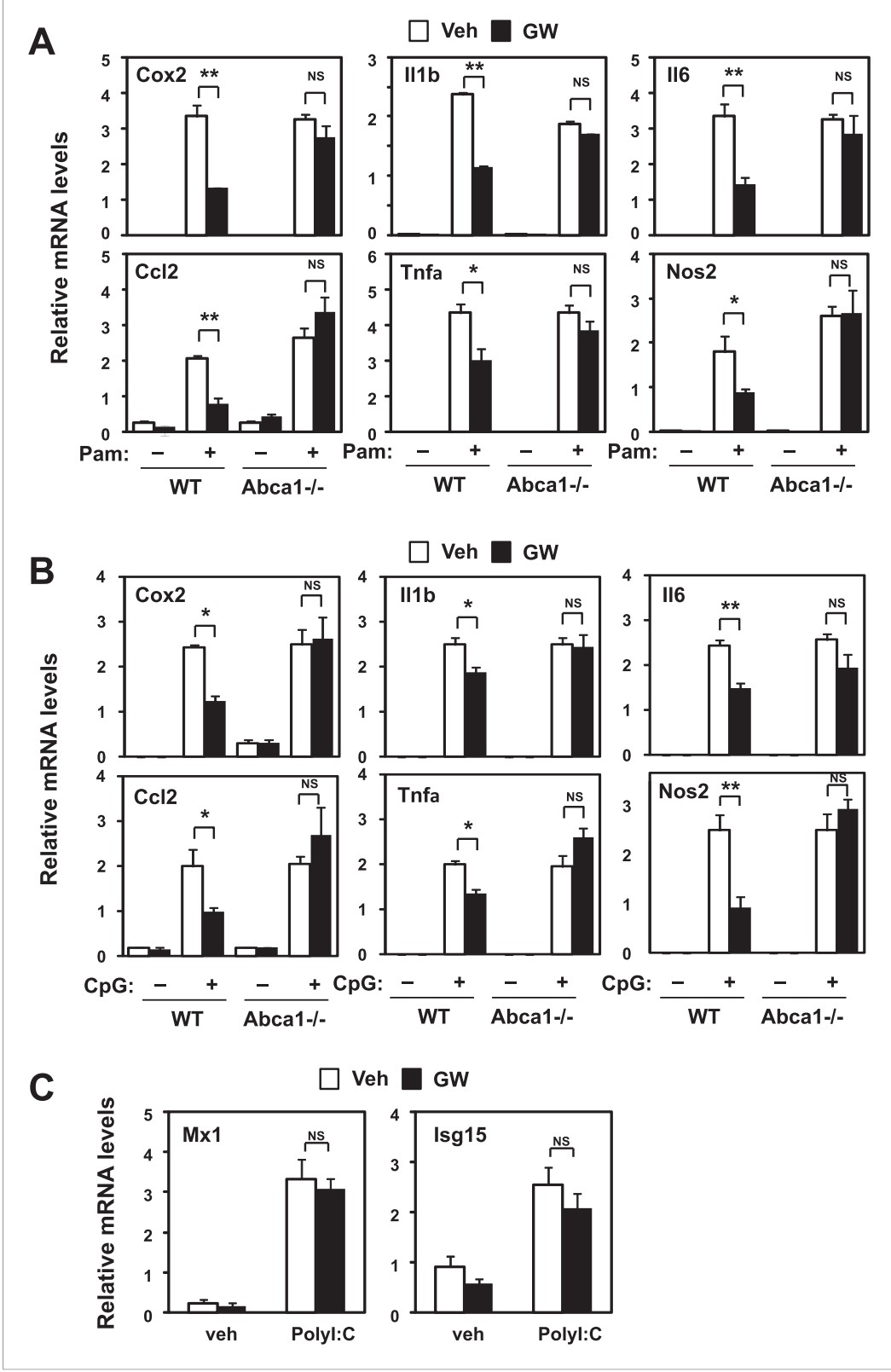

**Figure 10**. Inhibition of TLR2 and TLR9 signaling by LXR requires Abca1. Bone marrow-derived macrophages from myeloid-specific *Abca1*−/− and control wild-type mice (**A**, **B**, **C**) were pretreated with GW3965 (1 μM) overnight, followed by stimulation with Pam3CSK4 (100 ng/ml), LPS (10 ng/ml), CpG (1 μM) or polyI:C (10 μg/ml) for 4 hr as indicated. Gene expression was analyzed by real-time PCR. N = 4 per group. *p < 0.05, **p < 0.01, NS, not significant. Error bars represent means ± SEM.

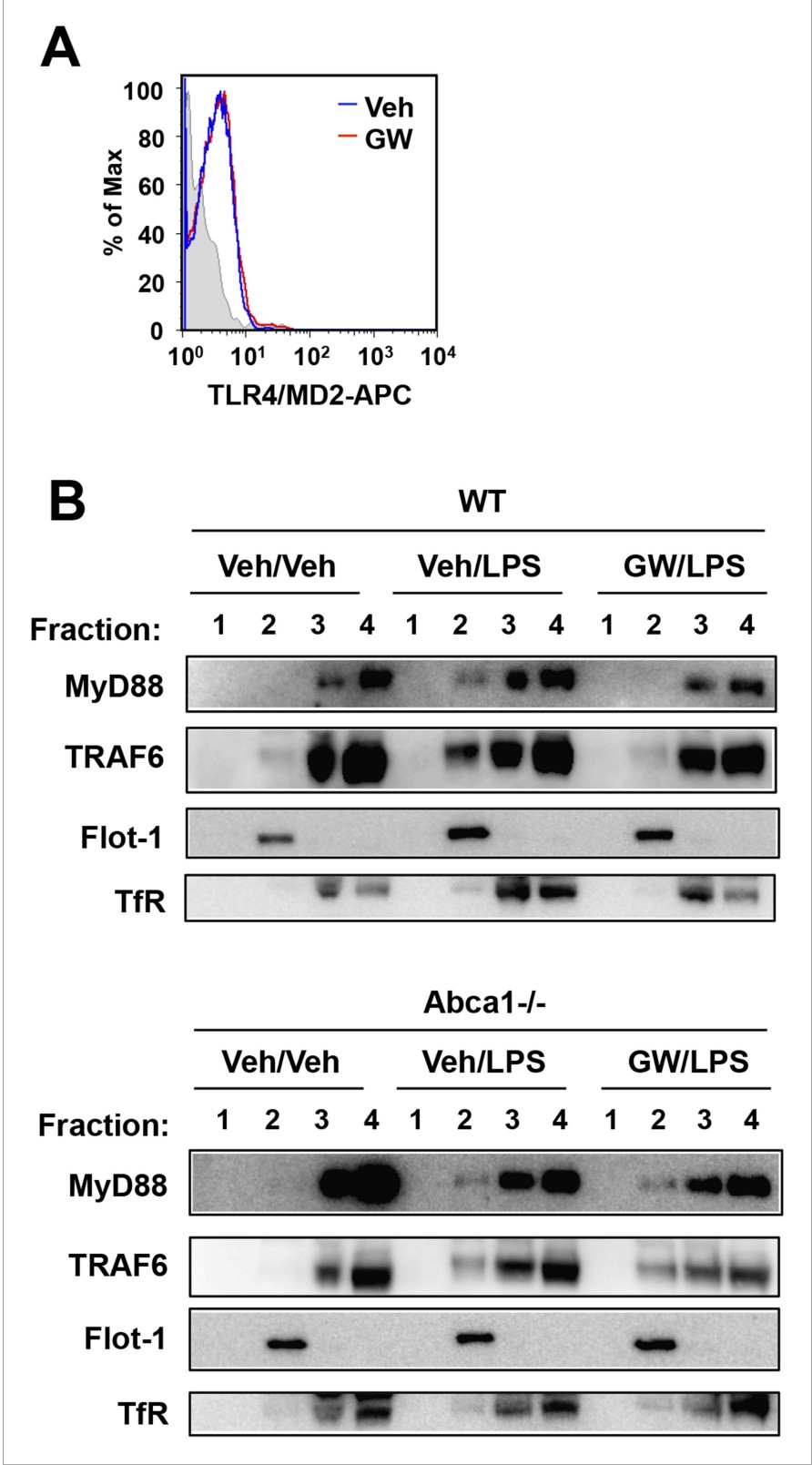

**Figure 11**. LXR activation inhibits recruitment of MyD88 and TRAF6 to lipid rafts. (**A**) Wild-type iBMDM were pretreated with GW3965 (1 μM) overnight and TLR4 expression was analyzed by flow cytometry. (**B**) Wild-type or Abca1−/− iBMDM were pretreated with GW3965 (1 μM) overnight, followed by stimulation with LPS (10 ng/ml) for 10

*Figure 11. continued on next page*

*Figure 11. Continued*

min. Lipid raft and non-raft fractions were isolated using detergent method and each fraction was analyzed by Western blotting. Results are representative of two independent experiments.

The following figure supplement is available for figure 11:

**Figure supplement 1**. Specificity of MyD88 and TRAF6 antibodies.

Previous studies have reported that *Abca1*-deficient macrophages contain more cholesterol and TLR4 protein in plasma membrane lipid rafts and that this results in enhanced inflammatory signaling upon LPS treatment (*Koseki et al., 2007*; *Zhu et al., 2008*, *2010*). Our data are consistent with these results and provide an additional mechanism whereby changes in Abca1 expression can affect TLR signaling. We found that Abca1 expression is important for the ability of LXR to inhibit signaling from TLRs 2, 4 and 9–all of which use MyD88 as a key adaptor. We showed that LXR activation inhibits recruitment of the adaptor proteins MyD88 and TRAF6 to rafts and thereby blocks TLR-induced activation of MAP kinases and NF-κB.

LXR activation leads to the induction of an entire cascade of genes involved in cellular lipid homeostasis (*Calkin and Tontonoz, 2012*; *Hong and Tontonoz, 2014*). Although Abca1 is clearly a major contributor, our data do not exclude the involvement of other factors. In our view, it is likely that additional LXR target genes involved in lipid handling may also contribute to effects on inflammation. One attractive additional player is the phospholipid-remodeling enzyme Lpcat3, which we have recently shown affects inflammatory signaling in hepatocytes in the setting of obesity (*Rong et al., 2013*).

In conclusion, these studies outline a mechanistic connection between regulated plasma membrane cholesterol distribution and TLR-mediated inflammatory signaling. They further support a unified view of the role of LXR in transcriptional regulation in which direct activation underlies the dual biological functions of these nuclear receptors in metabolism and inflammation.

## Materials and methods

### Experimental procedures

#### Reagents and plasmids

Synthetic LXR ligand GW3965 was provided by T. Wilson (GlaxoSmithKline). Human LXRα, LXRβ and their mutants were cloned into pDEST retrovirus vector using gateway technology (Invitrogen, Carlsbad, CA). Point mutations were introduced using the Quickchange site-directed mutagenesis kit (Stratagene) and verified by DNA sequencing.

#### Cell culture

Primary peritoneal macrophages were obtained from thioglycollate-treated mice 4 days after injection. For bone marrow-derived macrophages, bone marrow was isolated from femurs and tibias, and differentiated in DMEM supplemented with 20% fetal bovine serum (FBS), 30% L929 conditioned medium and antibiotics for 6–7 days. MEFs were immortalized by the SV40 Large T antigen retrovirus and selected with puromycin. Immortalized bone marrow-derived macrophages were obtained as previously described (*Blasi et al., 1985*; *Gandino and Varesio, 1990*). To reconstitute immortalized bone marrow derived macrophages from LXR-deficient mice with LXR wild-type or mutants, retrovirus supernatants from Phoenix E cells transfected with the expression vectors were infected to immortalized macrophages. The iBMDM stably expressing wild-type LXRα, LXRβ or mutants were obtained by 300 µg/ml of hygromycin B (Invitrogen). For RNAi experiments in macrophages, cells were transfected with control or ON-TARGETplus SMARTpool siRNAs (25 nM, Dharmacon) targeted RXRα, RXRβ, Ubc9, Hdac4, ABCA1, ABCG1 and TRAF6 using Darmafect 4 (Dharmacon). Cells were used for experiments after 48 hr incubation and target gene knockdown was validated by real-time PCR and Western Blot. For macrophage inflammatory responses, cells were placed in DMEM containing 0.5% FBS, 5 µM simvastatin, 100 µM mevalonic acid and 1 µM GW3965 or DMSO overnight. Cells were then stimulated with 10 ng/ml LPS or vehicle. For cellular cholesterol modification, bone marrow-derived macrophages were placed in DMEM containing 0.5% FBS, 5 µM simvastatin plus 100 µM mevalonic acid for 4 hr,

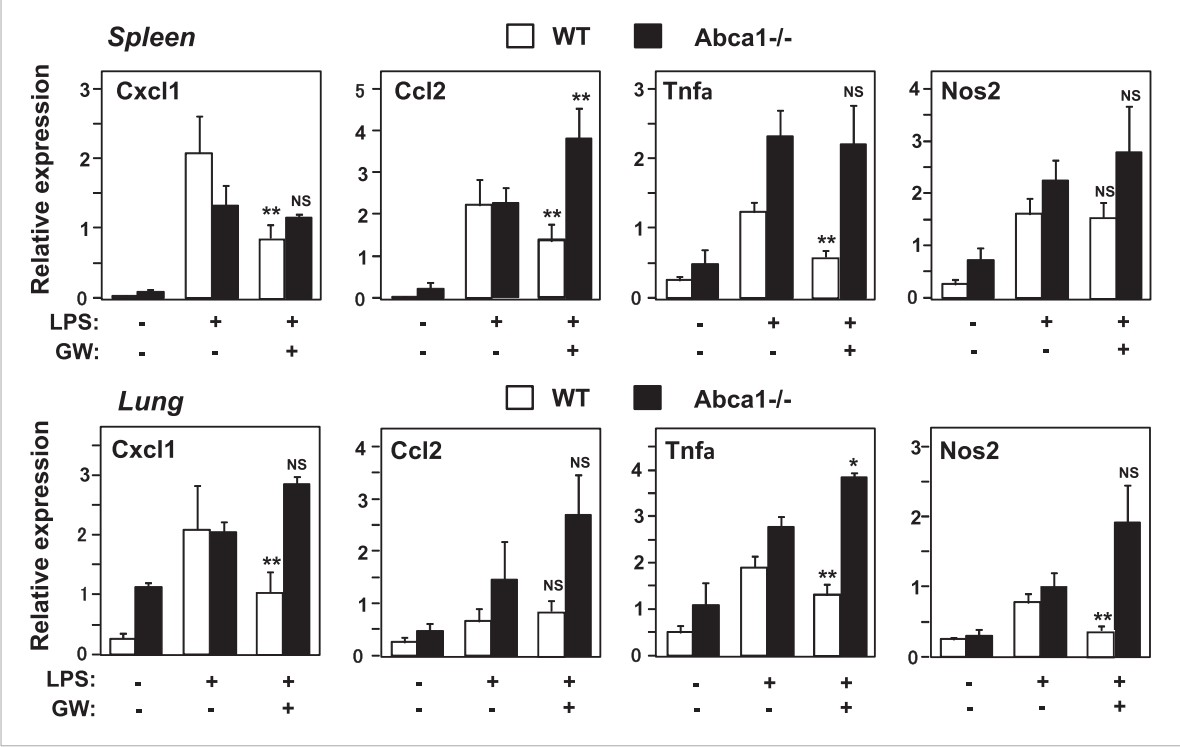

**Figure 12**. Abca1 contributes to LXR anti-inflammatory effects in vivo. Myeloid-specific *Abca1−/−* and control wild-type mice were gavaged with GW3965 (40 mg/kg) for 3 days, followed by challenge with LPS (1 mg/kg) by intraperitoneal injection. 2 hr later, spleens (upper) and lungs (lower) were harvested, RNA isolated and gene expression analyzed by real-time PCR. N = 3–4 per group. *p < 0.05, **p < 0.01, NS, not significant. Error bars represent means ± SEM.

then incubated with complexes of randomly methylated cyclodextrin (Trappsol) and cholesterol (Sigma–Aldrich) (CD/Chol, 100 µM) to overload with cholesterol (*Klein et al., 1995*) or hydroxypropyl-β-cyclodextrin (Trappsol) (CD, 10 mM) to deplete cholesterol for 1 hr and stimulated with LPS (10 ng/ml) for 4 hr.

## Gene expression and microarray analysis

Total RNA was isolated from cells and tissues with Trizol (Invitrogen), following the manufacturer's instruction. 500 ng of total RNA was used for cDNA synthesis and quantified by real-time PCR using SYBR Green (Diagenode) and an ABI 7900 instrument. Gene expression levels were normalized to 36B4. Primer sequences are given in *Table 1*. For microarray experiments, RNA was pooled from n = 3–4 biological replicates and processed in the UCLA Clinical Microarray Core Facility using SurePrint G3 Mouse GE 8 × 60 K Microarrays (Agilent). Data was analyzed using GenespringGX (Agilent). The data discussed in this publication have been deposited in NCBI's Gene Expression Omnibus (*Edgar et al., 2002*) and are accessible through GEO Series accession number GSE70444 (http://www.ncbi.nlm.nih.gov/geo/query/acc.cgi?acc=GSE70444) and GSE70463 (http://www.ncbi.nlm.nih.gov/geo/query/acc.cgi?acc=GSE70463).

## Chromatin immunoprecipitation (ChIP)

ChIP experiment was performed as described (*Villanueva et al., 2011*). Cells were crosslinked in DSG (disuccinimidyl glutarate) and 1% formaldehyde for 10 min and quenched with 2 M Glycine. Lysed cells were sonicated using a Bioruptor (Diagenode) according to the manufacturer's protocol, and chromatin was immunoprecipitated with antibodies against p65 (Santa Cruz C-20) or IgG (PP64, Millipore) overnight at 4°C for overnight in the presence of Protein A beads with salmon sperm DNA (Millipore). DNA enrichment was quantified by real-time PCR using SYBR Green (Diagenode) and an ABI 7900 instrument. Occupancy was normalized to input DNA. Primer sequences are given in *Table 2*.

**Table 1**. qPCR primers

| Murine qPCR primers | Forward | Reverse |
| --- | --- | --- |
| 36B4 | GGCCCTGCACTCTCGCTTTC | TGCCAGGACGCGCTTGT |
| Abca1 | CGTTTCCGGGAAGTGTCCTA | GCTAGAGATGACAAGGAGGATGGA |
| Abcg1 | TCACCCAGTTCTGCATCCTCT | GCAGATGTGTCAGGACCGAGT |
| Apoe | GACTTGTTTCGGAAGGAGCTG | CCACTCGAGCTGATCTGTCA |
| Cox-2 | CAGGTCATTGGTGGAGAGGTG | GGATGTGAGGAGGGTAGATCA |
| Cxcl1 | CACTGCACCCAAACCGAAGT | GGACAATTTTCTGAACCAAGGG |
| Hmgcs | GCCGTGAACTGGGTCGAA | GCATATATAGCAATGTCTCCT |
| Il-1b | AGAAGCTGTGGCAGCTACCTG | GGAAAAGAAGGTGCTCATGTCC |
| Il-6 | GCTACCAAACTGGATATAATCAGGA | CCAGGTAGCTATGGTACTCCAGAA |
| iNos | GCAGCTGGGCTGTACAAA | AGCGTTTCGGGATCTGAAT |
| Isg15 | CAGGACGGTCTTACCCTTTC | CGCTGCAGTTCTGTACCACT |
| Ldlr | AGGCTGTGGGCTCCATAGG | TGCGGTCCAGGGTCATCT |
| Mcp-1 | CATCCACGTGTTGGCTCA | GATCATCTTGCTGGTGAATGA |
| Mx-1 | AAACCTGATCCGACTTCACTTCC | TGATCGTCTTCAAGGTTTCCTTGT |
| Rxra | ACCGCTCCATAGCTGTGAAAG | TGAGCGCTGTTCCGGTGTA |
| Rxrb | GCCACTGGCATGAAAAGG | ATCTCCATCCCCGTCTTTGT |
| Srebp-1c | GGAGCCATGGATTGCACATT | GGCCCGGGAAGTCACTGT |
| Tnfa | TCTTCTCATTCCTGCTTGTGG | GGTCTGGGCCATAGAACTGA |
| **Human qPCR primers** | **Forward** | **Reverse** |
| 36B4 | CCACGCTGCTGAACATGCT | TCGAACACCTGCTGGATGAC |
| Abca1 | GCCTGCTAGTGGTCATCCTG | CCACGCTGGGATCACTGTA |
| Abcg1 | ATGTCAGGTATGGGTTCGAAG | TCTGGTCGATGTCACAGTGC |
| Cxcl1 | TCAAGAATGGGCGGAAAGC | CAGCATCTTTTCGATGATTTTC |
| IL-6 | CCAGGAGCCCAGCTATGAAC | CCCAGGGAGAAGGCAACTG |
| Lxra | AAGCCCTGCATGCCTACGT | TGCAGACGCAGTGCAAACA |
| Lxrb | TCGTGGACTTCGCTAAGCAA | GCAGCATGATCTCGATAGTGGA |
| Mcp-1 | AGAAGCTGTGATCTTCAAGACCATT | TGCTTGTCCAGGTGGTCCAT |
| Tnfa | TCTTCTCGAACCCCGAGTGA | CCTCTGATGGCACCACCAG |

## Protein analysis

Whole cell lysate were prepared in RIPA buffer (Boston Bioproducts) supplemented with protease inhibitors (Roche Diagnostics). The nuclear fractions were prepared as described (*Villanueva et al., 2011*). Proteins were separated on Nupage Bis-Tris gels then transferred to PVDF membrane (GE Osmonics).

Membranes were probed with antibodies against ABCA1 (Novus; NB400-105), LXRα (Perseus Proteomics; PP-K8607-10), LXRβ (Perseus Proteomics; PP-K8917-10), ERK (Cell Signaling; 9102), phospho-ERK (Cell Signaling; 9106), p38 MAPK (Cell Signaling; 9212), phospho-p38 MAPK (Cell

**Table 2**. ChIP primers

| Murine ChIP primers | Forward | Reverse |
| --- | --- | --- |
| Hbb2 | AGGTGCACCATGATGTCTGT | AGCAGGGTCAGTTGCTTCTT |
| Il-1b | GGACAATTGTGCAGATGGTG | CCTACCTTTGTTCCGCACAT |
| iNos | GGAGTGTCCATCATGAATGAG | CAACTCCCTGTAAAGTTGTGACC |
| Mcp-1 | TCCAGGGTGATGCTACTCCT | AGTGAGAGTTGGCTGGTGCT |

Signaling; 9216), JNK (Santa Cruz; sc-372), phospho-JNK (Cell signaling; 4668), IκBα (Cell Signaling; 9242), phosphor- IκBα (Santa Cruz; sc-7977), p65 (Santa Cruz; sc-372), MyD88 (Cell Signaling; 4283), TRAF6 (Santa Cruz; sc-7221), tubulin (Calbiochem; CP06), Lamin A/C (Santa Cruz; sc-6215), Flottilin-1 (Cell Signaling; 3253) and transferrin receptor (Invitrogen; 13–6800). Immunoblots were developed HRP-conjugated secondary antibodies (Invitrogen) and chemiluminescence kit (GE healthcare). The signals were detected with Image Quant LAS4000 (GE healthcare) and quantified with Image Quant TL7.0 (GE healthcare).

## Immunofluorescence
Cells were fixed in cold 4% paraformaldehyde for 10 min and permeabilized in 0.1% Triton X-100 at room temperature for 5 min. After blocking with 5% normal goat serum and 1% BSA in PBS, cells were incubated with rabbit anti-NF-κB p65 (1:100, Santa Cruz C-20) antibody at 4°C for overnight. After washes in PBS, coverslips were incubated at room temperature for 1 hr with the secondary antibody, Alexa488-conjugated goat anti-rabbit IgG (1:2000, Invitrogen A-11008). After cells were washed, cells were mounted in the presence of ProLong Gold Antifade Reagent with 4′,6′-diamidino-2-phenylindole (DAPI; Invitrogen). Images were collected with an LSM 510 confocal laser-scanning microscope (Carl Zeiss).

## Detergent-resistant membrane isolation and cholesterol measurement
Detergent-resistant membrane was isolated as described (*Lingwood and Simons, 2007*). Briefly, cells were washed and harvested in TNE buffer (150 mM NaCl, 2 mM EDTA, 50 mM Tris–HCl, pH7.4), lysed by passing through a 25 gauge needle, and solubilized with 1% TritonX-100 at 4°C for 30 min. Lysates were mixed with Optiprep to form a 40% iodixanol bottom layer in the centrifuge tube, overlaid with a 30% iodixanol layer and a 0% (TNE buffer) layer, and centrifuged for 2 hr at 260,000×*g* in TLA100.2 rotor. Four fractions were collected from the top of the gradient and subjected to immunoblot analysis. Cholesterol concentration in each fraction was measured by Amplex Red Cholesterol Assay Kit (Invitrogen) and normalized to protein concentration.

## Flow cytometry
Wild-type iBMDM were blocked with anti-mouse CD16/32 antibody (BioLegend). Cells were then stained with Alexa488-conjugated Cholera Toxin B subunit (1:100, Sigma–Aldrich) to examine the amount of lipid raft and APC-conjugated TLR4/MD2 complex (MTS510, eBioscience) or isotype control to examine TLR4/MD2 cell surface expression in 0.1% BSA, 5 mM EDTA in PBS. After washing, cells were analyzed on LSRII (BD bioscience) with FlowJo software (Treestar).

## Membrane dynamics
Membrane dynamics was analyzed as described (*Golfetto et al., 2013*). Briefly, wild-type iBMDM were plated onto 35-mm Mattek glass-bottom dishes and placed in DMEM containing 0.5% FBS, 5 μM simvastatin plus 100 μM mevalonic acid and 1 μM GW3965 or DMSO overnight. Cells were incubated with 50 μM Laurdan (6-dodecanoyl-2-dimethylaminonaphthalene; Invitrogen) at 37°C for 30 min. Cells were then rinsed with PBS and new medium was added. Fluorescence-lifetime imaging microscopy (FLIM) and ratiometric GP data were acquired with a Zeiss LSM710 META laser scanning microscope coupled to a 2-photon Ti:Sapphire laser (Mai Tai, Spectra Physics, Newport Beach, CA) producing 80-fs pulses at a repetition of 80 MHz with two different filters: 460/80 nm for the blue channel and 540/50 nm for the green channel. Spectral data were processed by the SimFCS software (Laboratory for Fluorescence Dynamics). Plasma membrane fluidity was calculated with GP and number of pixel.

## Animal studies
All animals were housed in a temperature-controlled room under a 12-hr light/12-hr dark cycle and under pathogen-free conditions. 24–26 weeks old male myeloid-specific *Abca1*−/− and control wild-type mice were gavaged with GW3965 (40 mg/kg) for 3 days. 2 hr after the last GW injection, the mice were challenged with LPS (1 mg/kg) or vehicle by intraperitoneal injection. Mice were sacrificed 2 hr later and spleen and lung were harvested. Animal experiments were conducted in accordance with the UCLA Animal Research Committee (UCLA Protocols 1999-131 and 2003-166).

## Statistic analysis
Statistics were performed using ANOVA with post hoc tests to compare to the control group. $p < 0.05$ was considered statistically significant. Data was presented as means ± SE.

## Acknowledgements

We thank Joseph Witztum and Robert Hegele for the Tangier fibroblasts. AI was funded by an AHA Fellowship (11POST7390075) and the Fellowship of Astellas Foundation for Research on Metabolic Disorders. EJT was supported by K99 HL118161 and AHA 13BGIA17080038. This work was supported by grants HL066088, DK063491, HL119962, GM103540, and GM076516. PT is an Investigator of the Howard Hughes Medical Institute.

## Additional information

### Competing interests

PT: Reviewing editor, *eLife*. The other authors declare that no competing interests exist.

### Funding

| Funder | Grant reference | Author |
| --- | --- | --- |
| Howard Hughes Medical Institute | | Peter Tontonoz |
| National Institutes of Health | HL066088 | Peter Tontonoz |
| National Institutes of Health | HL118161 | Elizabeth J Tarling |
| National Institutes of Health | GM103540 | Enrico Gratton |
| National Institutes of Health | GM076516 | Enrico Gratton |
| American Heart Association | 11POST7390075 | Ayaka Ito |
| National Institutes of Health | DK063491 | Peter Tontonoz |

The funders had no role in study design, data collection and interpretation, or the decision to submit the work for publication.

### Author contributions

AI, Conception and design, Acquisition of data, Analysis and interpretation of data, Drafting or revising the article; CH, PNH, Conception and design, Acquisition of data, Analysis and interpretation of data; XR, Acquisition of data, Analysis and interpretation of data; XZ, Acquisition of data, Contributed unpublished essential data or reagents; EJT, Acquisition of data, Drafting or revising the article; EG, Conception and design, Analysis and interpretation of data; JP, Conception and design, Analysis and interpretation of data, Drafting or revising the article, Contributed unpublished essential data or reagents; PT, Conception and design, Analysis and interpretation of data, Drafting or revising the article

### Author ORCIDs

Peter Tontonoz, http://orcid.org/0000-0003-1259-0477

### Ethics

Animal experimentation: This study was performed in strict accordance with the recommendations in the Guide for the Care and Use of Laboratory Animals of the National Institutes of Health. All of the animals were handled according to approved institutional animal care and use committee (IACUC) protocols (#99-131 and 2003-166) of the University of California, Los Angeles.

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
