## [Decision Letter]

Thank you for submitting your work entitled “Liver X receptors repress inflammation through ABCA1-dependent modulation of NF-κB and MAP kinase signaling” for peer review at *eLife*. Your submission has been favorably evaluated by Tadatsugu Taniguchi (Senior editor) and two reviewers, one of whom is a member of our Board of Reviewing Editors.

The reviewers have discussed the reviews with one another and the Reviewing editor has drafted this decision to help you prepare a revised submission.

Reviewers’ comments:

This manuscript provides compelling data refuting several papers suggesting that the anti-inflammatory effects of LXR agonism are due to increased SUMOylation of LXR, which was reported to recruit a co-repressor complex to promoters of inflammatory genes in macrophages. The present work suggests that SUMOylation is not required and that instead the mechanism involves the upregulation of ABCA1 expression. The authors show that ABCA1 is required for LXR-dependent anti-inflammatory activity through its ability to alter membrane lipid composition and in turn signaling through TLR4. While the data demonstrating that ABCA1 is required for the anti-inflammatory action of LXR agonists is quite good, the experiments were less robust in ruling out a role for SUMOylation or co-repression. Since a major conclusion of this work is based on co-repression not being involved, this is something that would need to be fleshed out in more detail. In addition, the proposed mechanism of ABCA1 effect (Figure 7) needs further experimental clarification.

1) The previous paper by Ghisletti et al., which was the basis for most of the conclusions that SUMOylation of LXRs mediates the anti-inflammatory effects, postulated that HDAC4 caused ligand-dependent SUMOylation via Ubc9, which in turn recruited N-CoR to target promoters (e.g., the iNos promoter). This previous paper used cells transfected with various siRNAs and/or LXR mutants to support their conclusions. The present manuscript has a similar set of experiments using retroviral expression in LXR KO and ABCA1 KO cells to show that LXR mutants lacking the SUMO sites still repress some genes. However, these experiments cannot be directly compared to Ghisletti et al., since the present work never even investigates the requirement of Ubc9 or N-CoR. In addition, perhaps the discrepancy may be due to the promoters studied in each paper. Ghisletti mainly studied the iNos promoter, while this paper focused mostly on other targets (e.g., IL-1b). At this point I do not believe one can entirely rule out the co-repression as part of the mechanism. The involvement of Ubc9 and N-CoR should be investigated. It would be important to include iNOS genes in the analyses here, as the effects may well be gene dependent and so direct comparison impossible and the issue will remain unresolved unless the same genes are analyzed.

2) I note that the present paper as well as the previous papers used siRNA as well as overexpression of mutant proteins to support their conclusions. Therefore, please include data showing the level of expression of proteins introduced into cells by retroviral infection, and the level of knockdown achieved by siRNA treatment. These are important controls to show the physiologic relevance of the findings.

3) Why is the LPS responds in the LXR-null macrophages slower than it is in WT cells (Figure 5)? LPS induced Erk, p38 and JNK phosphorylation within 20 min in WT but not in the KO cells. It may be informative to also include a time course of this effect in the Abca1-null cells (Figure 5) as well. It should be the same as the LXR KOs.

4) Please include statistical analysis in Figure 6 to show which effects are significant.

5) Figure 7 needs statistical quantitation. Visual inspection of this figure does not reveal “markedly reduced” changes in any of the bands due to LXR agonism. It is also not clear why flot-1 is not detected in any fractions in Figure 7? This doesn't seem to correspond to what the authors are saying in the text. Please clarify. I found it hard to interpret Figure 7.

7) Figure 7 are equally uninterpretable: the result showing MyD88 recruitment between Raw cells and BMDMs looks opposite – in one case LPS increases the recruitment, in another reduces it. I feel this is not the best assay to address the mechanism. One likely scenario is that ABCA1 mediated cholesterol efflux affects recruitment of TIRAP. TIRAP is a sorting adaptor responsible for recruitment of MyD88 to TLR4 at the plasma membrane. TIRAP interacts with PIP2, which is enriched at lipid rafts. So a plausible scenario is that cholesterol 'depletion' from PM may affect TIRAP mediated MyD88 recruitment. A direct way to test this is by looking at TIRAP recruitment to PM in response to LPS (Kagan, Cell. 2006 Jun 2;125(5):943-55.). This is a fairly simple experiment that can be done quickly and all the reagents are available from the authors of the above study.

8) Recruitment of TIRAP-MyD88 should only affect NF-kB/MAPK pathways, but not the TRAM-TRIF pathways leading to IFN-stimulated genes (ISGs). Is ISG expression affected in the conditions tested in the study (GW, ABCA-1 KO)? The authors already have the data, just need to discuss it.

9) Further to point 3 above, the authors should test the effect of LXR and ABCA1 deficiency on CpG induced gene expression. Regardless of the outcome, the result of this simple experiment would be very informative.

---

## [Author Response]

*[…] The authors show that ABCA1 is required for LXR-dependent anti-inflammatory activity through its ability to alter membrane lipid composition and in turn signaling through TLR4. While the data demonstrating that ABCA1 is required for the anti-inflammatory action of LXR agonists is quite good, the experiments were less robust in ruling out a role for SUMOylation or co-repression. Since a major conclusion of this work is based on co-repression not being involved, this is something that would need to be fleshed out in more detail. In addition, the proposed mechanism of ABCA1 effect (*Figure 7*) needs further experimental clarification.*

*1) The previous paper by Ghisletti et al., which was the basis for most of the conclusions that SUMOylation of LXRs mediates the anti-inflammatory effects, postulated that HDAC4 caused ligand-dependent SUMOylation via Ubc9, which in turn recruited N-CoR to target promoters (e.g., the iNos promoter). This previous paper used cells transfected with various siRNAs and/or LXR mutants to support their conclusions. The present manuscript has a similar set of experiments using retroviral expression in LXR KO and ABCA1 KO cells to show that LXR mutants lacking the SUMO sites still repress some genes. However, these experiments cannot be directly compared to Ghisletti et al., since the present work never even investigates the requirement of Ubc9 or N-CoR. In addition, perhaps the discrepancy may be due to the promoters studied in each paper. Ghisletti mainly studied the iNos promoter, while this paper focused mostly on other targets (e.g., IL-1b). At this point I do not believe one can entirely rule out the co-repression as part of the mechanism. The involvement of Ubc9 and N-CoR should be investigated. It would be important to include iNOS genes in the analyses here, as the effects may well be gene dependent and so direct comparison impossible and the issue will remain unresolved unless the same genes are analyzed*.

This is an important point. In response to the reviewers’ comments, we have now performed additional knockdowns for Ubc9 and HDAC4 (Figure 2—figure supplement 2). The data indicate that neither of these factors is required for repression in our system. We have also included iNos gene expression in our analysis as requested in multiple Figures 2, 3, 8 and 10. iNos is not expressed by fibroblasts and therefore could not be included in the MEF studies. We have not performed N-CoR knockdowns, because this experiment is not informative in our view. Loss of N-CoR derepresses LXR target genes, and this actually leads to increased Abca1 expression, as well as a myriad of other effects on many different nuclear receptors.

We do not wish to claim that there is no role for SUMOylation in any context. However, our data do clearly show that it is possible for LXR to repress inflammation in the absence of SUMOylation. The SUMOylation hypothesis was primarily based on analysis artificial promoters and it was never tested in vivo. Our systems examine endogenous genes and, as outlined in more detail in point 2 below, does not employ supraphysiological levels of LXRs.

2) I note that the present paper as well as the previous papers used siRNA as well as overexpression of mutant proteins to support their conclusions. Therefore, please include data showing the level of expression of proteins introduced into cells by retroviral infection, and the level of knockdown achieved by siRNA treatment. These are important controls to show the physiologic relevance of the findings.

Thank you for this comment. We neglected to make it clear that we are not massively overexpressing LXRs in our system. Retroviral reconstitution of LXRs in the KO cells results in LXR protein expression within the physiological range and restores LXR target gene expression to levels comparable to WT cells (see Figure 1—figure supplement 1). We have now also provided additional data on protein expression for various mutant LXRs (Figure 1—figure supplement 1). We have also provided additional validation of the efficacy of the siRNA reagents (Figure 1—figure supplement 1, Figure 2—figure supplement 1). Note, the key experiments on Abca1 are all done with true KO cells and the absence of protein is confirmed by western (see Figure 6).

*3) Why is the LPS responds in the LXR-null macrophages slower than it is in WT cells (*Figure 5*)? LPS induced Erk, p38 and JNK phosphorylation within 20 min in WT but not in the KO cells. It may be informative to also include a time course of this effect in the Abca1-null cells (*Figure 5*) as well. It should be the same as the LXR KOs*.

This was an insightful comment. As requested we have now included the comparable time course for Abca1 in the new Figure 6. As predicted by the reviewer, the response is in fact similar to the LXR KOs. Thus, both knockouts share a common basal alteration in MAPK response, further emphasizing the link between LXR and Abca1 in inflammatory signaling.

*4) Please include statistical analysis in*
Figure 6
*to show which effects are significant.*

We have now provided statistical analysis and also expanded the ChIP analysis to include both WT and Abca1-/- cells in the revised Figure 8.

*5)*
Figure 7
*needs statistical quantitation. Visual inspection of this figure does not reveal* “*markedly reduced*” *changes in any of the bands due to LXR agonism. It is also not clear why flot-1 is not detected in any fractions in*
Figure 7*? This doesn't seem to correspond to what the authors are saying in the text. Please clarify. I found it hard to interpret*
Figure 7.

We have redone the studies in the original Figure 7 as outlined below and the results are now more clear. We apologize that the Flot-1 western image did not appear on the reviewer’s PDF. It did appear on our own PDF downloaded from the *eLife* site. We have remade the image and hope that this will correct the PDF conversion error (Revised Figure 9). Note, although the magnitude of change in the cholesterol levels in the raft fraction is not large in absolute terms, it is nonetheless biologically significant. Cholesterol levels in membranes never change dramatically in any physiological context. E.g., a change in ER cholesterol content between 3 and 5% has been reported by Brown and Goldstein to control SREBP cleavage.

*7)*
Figure 7
*are equally uninterpretable: the result showing MyD88 recruitment between Raw cells and BMDMs looks opposite – in one case LPS increases the recruitment, in another reduces it. I feel this is not the best assay to address the mechanism. One likely scenario is that ABCA1 mediated cholesterol efflux affects recruitment of TIRAP. TIRAP is a sorting adaptor responsible for recruitment of MyD88 to TLR4 at the plasma membrane. TIRAP interacts with PIP2, which is enriched at lipid rafts. So a plausible scenario is that cholesterol 'depletion' from PM may affect TIRAP mediated MyD88 recruitment. A direct way to test this is by looking at TIRAP recruitment to PM in response to LPS (Kagan, Cell. 2006 Jun 2;125(5):943-55.). This is a fairly simple experiment that can be done quickly and all the reagents are available from the authors of the above study*.

We have repeated and refined the assays in the old Figure 7. We agree that the RAW cell data was ambiguous, and we now present data exclusively from bone marrow macrophages, as they represent the most physiological system. We now compare WT and ABCA1-/- macrophages and show that LXR activation clearly inhibits MyD88 and TRAF6 recruitment to rafts in WT but not Abca1-/- cells (new Figure 11). We also appreciate the insightful suggestion regarding TIRAP, but based on our finding that TLR9 is also inhibited by LXR, TIRAP seems to be excluded as the mediator of the LXR effect.

*8) Recruitment of TIRAP-MyD88 should only affect NF-kB/MAPK pathways, but not the TRAM-TRIF pathways leading to IFN-stimulated genes (ISGs). Is ISG expression affected in the conditions tested in the study (GW, ABCA-1 KO)? The authors already have the data, just need to discuss it*.

We have now provided and discussed this data as requested (Figure 10). LXRs do not repress activation of the TRAM-TRIF-IRF3 arm stimulated by TLR3, consistent with MyD88 being the target.

*9) Further to point 3 above, the authors should test the effect of LXR and ABCA1 deficiency on CpG induced gene expression. Regardless of the outcome, the result of this simple experiment would be very informative*.

This was an excellent suggestion. We have now examined LXR effects on signaling through TLR2,3 and 9 (Figure 10). LXR represses TLR9 in an Abca1-dependent manner, again suggesting that MyD88 rather than TIRAP is the target.